# Defective lytic transglycosylase disrupts cell morphogenesis by hindering cell wall de-*O*-acetylation in *Neisseria meningitidis*

Allison Hillary Williams[1]*, Richard Wheeler[1,2], Ala-Eddine Deghmane[3], Ignacio Santecchia[1,4], Ryan E Schaub[5], Samia Hicham[1], Maryse Moya Nilges[6], Christian Malosse[7], Julia Chamot-Rooke[7], Ahmed Haouz[8], Joseph P Dillard[5], William P Robins[9], Muhamed-Kheir Taha[3], Ivo Gomperts Boneca[1]*

[1]Unité Biologie et Génétique de la Paroi Bactérienne, Institut Pasteur; Groupe Avenir, INSERM 75015, Paris, France; [2]Tumour Immunology and Immunotherapy, Institut Gustave Roussy, Villejuif, France; [3]Unité des Infection Bactériennes Invasives, Institut Pasteur, Paris, France; [4]Université Paris Descartes, Sorbonne Paris Cité, Paris, France; [5]Department of Medical Microbiology and Immunology, University of Wisconsin-Madison, Madison, United States; [6]Unité Technologie et Service BioImagerie Ultrastructural, Institut Pasteur, Paris, France; [7]Unité Technologie et Service Spectrométrie de Masse pour la Biologie, Institut Pasteur; UMR 3528, CNRS 75015, Paris, France; [8]Plate-forme de Cristallographie-C2RT, Institut Pasteur; UMR3528, CNRS 75015, Paris, France; [9]Department of Microbiology, Harvard Medical School, Boston, United States

*For correspondence:
awilliam@pasteur.fr (AHW);
bonecai@pasteur.fr (IGB)

Competing interests: The authors declare that no competing interests exist.

**Abstract** Lytic transglycosylases (LT) are enzymes involved in peptidoglycan (PG) remodeling. However, their contribution to cell-wall-modifying complexes and their potential as antimicrobial drug targets remains unclear. Here, we determined a high-resolution structure of the LT, an outer membrane lipoprotein from *Neisseria* species with a disordered active site helix (alpha helix 30). We show that deletion of the conserved alpha-helix 30 interferes with the integrity of the cell wall, disrupts cell division, cell separation, and impairs the fitness of the human pathogen *Neisseria meningitidis* during infection. Additionally, deletion of alpha-helix 30 results in hyperacetylated PG, suggesting this LtgA variant affects the function of the PG de-*O*-acetylase (Ape 1). Our study revealed that Ape 1 requires LtgA for optimal function, demonstrating that LTs can modulate the activity of their protein-binding partner. We show that targeting specific domains in LTs can be lethal, which opens the possibility that LTs are useful drug-targets.

## Introduction

Lytic transglycosylases (LTs) degrade peptidoglycan (PG) to produce *N*-acetylglucosamine (GlcNAc)−1,6-anhydro-*N*-acetylmuramic acid (MurNAc)-peptide (G-anhM-peptide), a key cytotoxic elicitor of harmful innate immune responses (*Viala et al., 2004*). LTs have been classified into four distinct families based on sequence similarities and consensus sequences. LTs belonging to family 1 of the glycoside hydrolase (GH) family 23 share sequence similarity with the goose-type lysozyme (*Blackburn and Clarke, 2001*). Family 1 can be further subdivided into five subfamilies, 1A through E, which are all structurally distinct (*Blackburn and Clarke, 2001*). Despite the overall structural differences among LTs, their active sites, enzymatic activities and substrate specificities are fairly well conserved.

**eLife digest** Bacteria are surrounded by a tough yet flexible wall that protects the cell and serves as an anchor for several of the cell's structures. This cell wall contains a large mesh-like molecule called peptidoglycan made of many repeated building blocks. When a bacterial cell divides in two, it needs to make more of this material. Making peptidoglycan involves two different sets of enzymes working together: "polymerases" are the enzymes that link the individual building blocks to peptidoglycan, one after the other; while "lytic transglycosylases" are enzymes that modify the peptidoglycan to create space for the addition of new building blocks and for assemblies of proteins that must span the cell wall.

Lytic transglycosylases are known to assemble with other proteins and enzymes to form the cell's peptidoglycan-modifying machinery, but it was not clear exactly what purpose they serve within these "enzyme complexes". It was also unclear whether these enzymes would be good targets for new antibiotics.

To help answer these questions, Williams et al. looked at a lytic transglycoslyase called LtgA. This enzyme is originally from *Neisseria meningitidis*, a bacterium that can cause meningitis and life-threatening sepsis in humans. Williams et al. discovered that part of the enzyme's active site – the region of an enzyme where the chemical reaction takes – can switch from an ordered helix to a disordered, flexible loop. Bacteria were then genetically engineered to make a version of the enzyme that lacked this helix. These bacteria had weaker cell walls and were deformed; they were also less able to grow and divide, both in the laboratory and in a mouse model of infection. Further analysis showed that the deletion of the helix from the enzyme resulted in the peptidoglycan being modified much more than normal, which could likely explain their reduced virulence.

Williams et al. also found that deleting the helix from LtgA interfered with the activity of a protein that interacts with this enzyme, called Ape1, which also contributed to the fragility of the cell wall. This shows that lytic transglycosylases assembled into enzyme complexes can alter the activities of other proteins in the complex.

Together these findings show that researchers could target one enzyme in a complex in bacteria, and disrupt the activity of other proteins in that complex. This highlights the possibility of considering enzyme complexes as useful targets for new drugs, which is important considering the current problem of antibiotic resistance.

The crystal structure of the outer membrane lipoprotein LtgA, a homolog of Slt70 that belongs to family 1A of GH family 23 from the pathogenic *Neisseria* species, was previously determined at a resolution of 1.4 Å (*Figure 1a*). (*Williams et al., 2017*; *Williams et al., 2018*). Briefly, LtgA is a highly alpha-superhelical structure consisting of 37 alpha helices (*Figure 1a*). Although LTs have very diverse overall secondary structures, they exhibit similar substrate specificities and a preference for PG (*Vollmer et al., 2008*). LtgA shares an overall weak sequence similarity with Slt70 (25%). However, the structural and sequence alignments of the catalytic domains of Slt70 and LtgA revealed absolute active site conservation (*Williams et al., 2018*). The active site of LtgA is formed by ten alpha helices ($\alpha$ 28, 29, 30, 31, 32, 33, 34, 35, 36, 37), with a six-alphahelix bundle ($\alpha$ 29, 30, 31, 32, 33, 34) constituting the core of the active site that firmly secures the glycan chain (*Figure 1a*).

LTs utilize a single catalytic residue, either a glutamate or aspartate, which plays the role of an acid and then that of a base (*Thunnissen et al., 1994*; *van Asselt et al., 1999*; *Scheurwater et al., 2008*; *Reid et al., 2004*; *van Asselt and Dijkstra, 1999*). In our recent study, active LtgA was monitored for the first time in the crystalline state, and the residues involved in the substrate and product formation steps were identified. Globally, conformational changes occurred in three domains, the U, C and L domains, between native LtgA and LtgA bound to the product (*Williams et al., 2018*). Substantial conformational changes were observed in the active site, for example, during the product formation step, the active site adopted a more open conformation (*Williams et al., 2018*).

Many Gram-negative bacteria have multiple and redundant LTs; for example, *Escherichia coli* has eight (MltA, MltB, MltC, MltD, MltE, MltF, MltG and Slt70), and *Neisseria* species encode 5 (LtgA, LtgB, LtgC, LtgD, and LtgE). Because the activity of LTs is redundant, the loss of one or more LTs in *E. coli* leads to no observable growth defects. When genes for six LTs were deleted from *E. coli*, a

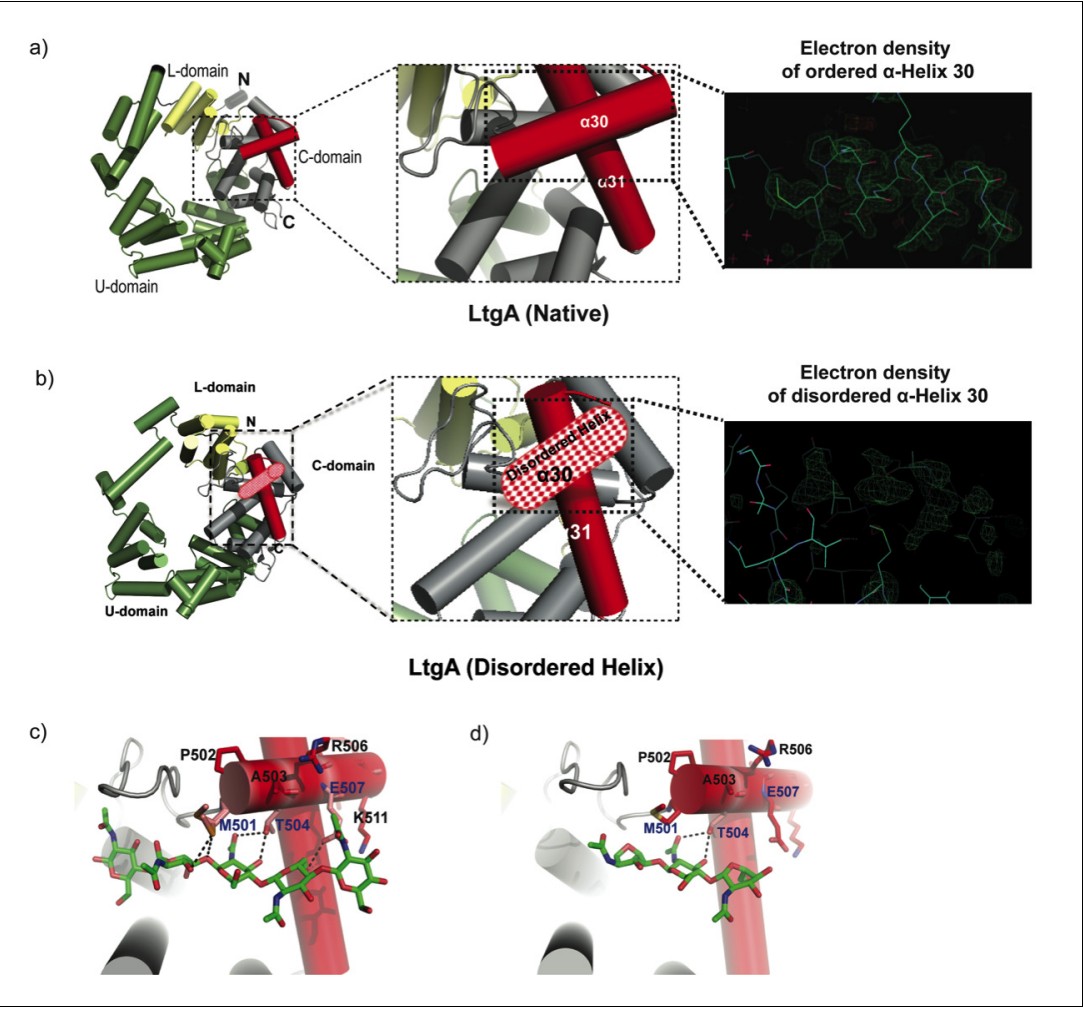

**Figure 1.** Molecular architecture of LtgA alpha helix 30 and contacts made with reaction. intermediates. (**a**) Native structure of LtgA. Ribbon model of LtgA displaying a helical structure consisting of 37 alpha helices. LtgA consists of three domains: A C-domain (gray and red), which houses the putative catalytic domain, and the L (yellow) and U (green) domains, which are of unknown function. A long N-terminal extension interacts with the L-domain, which closes the structure (PDB ID: 5O29). Clear and consistent density for helix 30 was depicted by the Fo-Fc omit map (green) (**b**) LtgA with a disordered conformation of helix 30. Clear and consistent density for helix 30 was absent as depicted by the Fo-Fc omit map (green) of helix 30 (PDB ID: 6H5F). (**c**) LtgA plus trapped intermediates (chitotetraose and a GlcNAc sugar) (PDB ID: 5O2N). (**d**) LtgA plus anhydro product (1,6-anhydro-chitotriose) (PDB ID: 5OIJ).

The online version of this article includes the following figure supplement(s) for figure 1:

**Figure supplement 1.** Conservation of alpha helix 30 amongst diverse lytic transglycosylases.

**Figure supplement 2.** Binding of LtgA to the Peptidoglycan.

---

mild chaining phenotype was observed (*Heidrich et al., 2002*). However, despite lack of strong observable phenotypic changes, it has been suggested that LTs may have well-defined roles in the cell. For example, the deletion of *ltgA* and *ltgD* in *Neisseria gonorrhoeae* eliminates the release of cytotoxic PG monomers suggesting the activities of LtgA and LtgD are redundant. Moreover, LtgA primarily localizes at the septum, indicating a role in the divisome machinery, whereas LtgD is distributed along the entire cell surface (*Schaub et al., 2016*).

The activities of LTs are known to be inhibited by β-hexosaminidase inhibitors (for example, NAG-thiazoline); by bulgecins A, B and C; and by PG-O-acetylation (*Williams et al., 2017*; *Reid et al., 2004*; *Templin et al., 1992*; *Tomoshige et al., 2018*). PG-O-acetylation (*Weadge et al.,*

*2005*) is a process that allows pathogenic bacteria to subvert the host innate immune response (*Diacovich and Gorvel, 2010*; *Aubry et al., 2011*). It should be noted that many Gram-positive and Gram-negative bacteria *O*-acetylate their PG, with a few notable exceptions such as *E. coli* and *Pseudomonas aeruginosa* (*Clarke et al., 2010*). Peptidoglycan *O*-acetylation prevents the normal metabolism and maturation of PG by LTs (*Bera et al., 2005*). Ape1, a PG de *O*-acetylase, is present in *Neisseria* species and generally in Gram-negative bacteria that *O*-acetylate their PG. Ape1 catalyzes the hydrolysis of the *O*-acetyl modification specifically at the sixth carbon position of the muramoyl residue, thus assuring the normal metabolism of PG by LTs (*Weadge et al., 2005*; *Weadge and Clarke, 2006*; *Pfeffer and Clarke, 2012*).

LTs form protein complexes with other members of the PG biosynthetic apparatus, such as PBPs (*Vollmer et al., 2008*; *Dijkstra and Thunnissen, 1994*; *Romeis and Höltje, 1994*; *van Heijenoort, 2011*; *Legaree and Clarke, 2008*; *Vollmer and Bertsche, 2008*). Most notable are the interactions between Slt70 and PBPs 1b, 1c, 2 and 3 (*von Rechenberg et al., 1996*). PBPs are essential for bacterial cell wall synthesis and are required for proliferation, cell division and the maintenance of the bacterial cell structure. Previously, PBPs were thought to be primarily responsible for the polymerization of PG. Recently, RodA, a key member of the elongasome, and a shape, elongation, division and sporulation (SEDS) protein family member was shown to be a PG polymerase. RodA functions together with PBP2 to replicate the transglycosylase and transpeptidase activities found in bifunctional PBPs (*Cho et al., 2016*; *Meeske et al., 2016*; *Sjodt et al., 2018*). SEDS proteins are widely distributed in bacteria and are important in both the cell elongation and division machinery. *Neisseria* species such as *N. gonorrhoeae* and *N. meningitidis* are coccoid in shape and lack an elongation machinery. Therefore, these species incorporate new PG through complex interactions in the divisome. Both *N. gonorrhoeae* and *N. meningitidis* have five PBPs, namely, PBP1, PBP2, PBP3, PBP4 and PBP5. PBP1 and PBP2 are homologous to *E. coli* PBP1a and PBP3, while the *Neisseria* PBP3 and PBP4 are homologous to *E. coli* PBP4 and PBP7 (*Sauvage et al., 2008*). PBP5 in both *E. coli* and *Neisseria* species are both predicted carboxypeptidases (*Zarantonelli et al., 2013*). FtsW, a RodA homolog and a key component of the divisome machinery, forms a complex with FtsI (PBP3). The FtsW-PBP3 complex shares similar interacting regions with the RodA-PBP2 complex, and is the confirmed PG polymerase of the divisome (*Taguchi et al., 2019*).

Previous work by our group and others have demonstrated that PBPs and LTs can be targeted in a combined antibiotic regimen that could counter antibiotic resistance (*Bonis et al., 2012*), highlighting the possibility of simultaneously inhibiting LTs and their binding partners, such as PBPs, to achieve a synergistic antibiotic effect. Here, we reveal the near-atomic-resolution crystal structure of a native version of LtgA with a disordered active site alpha helix. When LtgA, missing the alpha helix 30 motif, was expressed from an ectopic locus in *N. meningitidis* (at an elevated level compared to wild type), bacterial growth, cell division and daughter cell separation were disrupted, compromising the integrity of the cell wall and PG composition, and diminishing bacterial fitness or virulence in a mouse infection model. It is known that LTs exist in multi-protein complexes. Here, we demonstrate that LTs can enhance the activity of one of their protein-binding partners thus ascribing a new role to LTs in the PG degrading machinery. This study demonstrates that despite the redundancy of LTs, they can be useful potential targets for future antibiotic development.

## Results

### Structure of LtgA with a disordered alpha helix 30

In the course of monitoring the LtgA reaction in the crystalline state, we captured a native version of LtgA with a distinctly disordered alpha helix 30 (*Figure 1a–b*, *Video 1*). This represents a newly identified conformational state of LtgA whereby alpha helix 30 transitions from an ordered to a disordered state (*Figure 1a–b*).

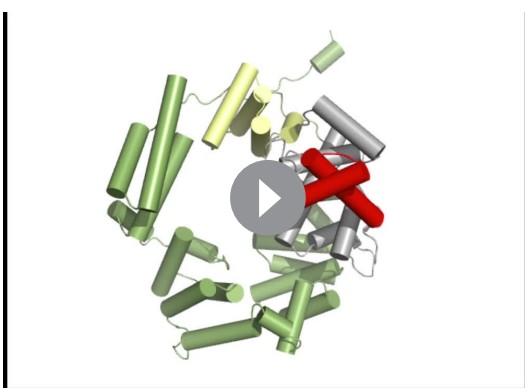

**Video 1.** Video of LtgA cleaving a glycan strand.
https://elifesciences.org/articles/51247#video1

Interestingly, this disorder of alpha helix 30 did not affect the overall structural integrity of the active site (*Figure 1b*, *Video 1*) because all the other helices making up the catalytic domain remained intact. Moreover, LtgA was already shown to be active in the crystalline state in our previous studies, although the molecular details of alpha helix 30 interactions with the ligand was not addressed (*Figure 1a–b*, *Video 1*; *Williams et al., 2018*).

Alpha helix 30, with the sequence [501](MPATAREIAGKIGMD)[516] (*Figure 1a–b*, colored in light red), is structurally conserved among the closest homologs of LtgA, mainly, Slt's, and other LTs such as MltE and MltC (*van Asselt et al., 1999*; *Artola-Recolons et al., 2011*; *Fibriansah et al., 2012*; *Artola-Recolons et al., 2014*; *Höltje, 1996*; *Figure 1—figure supplement 1*). Alpha helix 30 clamps the glycan strand during catalysis (*Figure 1c*, *Video 1*) and undergoes conformational changes to a more open conformation after product formation (*Figure 1d*). Met 501 and Glu 507 of alpha helix 30 lose hydrogen-bonding contact with the ligand after product formation (*Figure 1d*, *Video 1*). Consistent with the structural data showing the role of alpha helix 30 in substrate/product binding, a heterologously expressed and purified LtgA$^{\Delta30}$ showed severely diminished PG-binding capabilities when compared to wild-type LtgA or mutants of residues involved in the catalytic mechanism or substrate binding (E481A, E580) of LtgA (*Figure 1—figure supplement 2*). This further emphasizes the potential critical structural role of alpha helix 30 in the function of LtgA and consequently in the proper metabolism of the PG.

## The functional role of alpha helix 30

Given the important structural role of LtgA alpha helix 30, we investigated its functional role in vivo by engineering the following constructs in *N. meningitidis*: i) an LtgA knockout strain ($\Delta ltgA$), ii) a knockout strain complemented at an ectopic locus on the meningococcal chromosome with the wild-type gene ($\Delta ltgA^{ltgA}$), or iii) complemented with alpha helix 30 deletion ($\Delta ltgA^{ltgA\Delta30}$). Similar to other LTs, a complete deletion of the *ltgA* gene from the chromosome did not affect the growth of the bacteria (*Figure 2a*; *Chan et al., 2012*). Interestingly, the strain with *ltgA* lacking the alpha helix 30 coding sequence exhibited severely reduced growth (*Figure 2a*), with an exponential phase growth rate (0.059 h$^{-1}$ ±0.012) significantly different from that of the wild-type or $\Delta ltgA^{ltgA}$ strain (0.72 h$^{-1}$ ±0.15 or 0.21 h$^{-1}$ ±0.043, respectively) based on the calculated slopes of the growth curves (p<0.0001) (*Figure 2a*).

To exclude concerns about LtgA stability and to confirm that LtgA$^{\Delta30}$ continued to be expressed, the degradation of LtgA across all four strains was examined by western blot of lysates of bacteria harvested at various time points after incubation with chloramphenicol (*Figure 2b–c*). As expected, LtgA was not detected in the $\Delta ltgA$ knockout mutant (*Figure 2b–c*). The levels of LtgA or LtgA$^{\Delta30}$ in the $\Delta ltgA^{ltgA}$ and $\Delta ltgA^{ltgA\Delta30}$ strains was 4.2 and 3.7 times higher, respectively, than observed in the wild-type strain at $t_0$, possibly because the transcription of *ltgA* was controlled by a stronger promoter in these strains when compared to the parental strain. After the addition of chloramphenicol, LtgA appeared to be maintained at comparable levels in the wild-type and complemented strains, and the levels decreased slowly during the sampling period, as revealed by quantitative measurement of relative protein abundance using densitometry ($t_{1/2}$ > 9 h) (*Figure 2b–c*).

The promoter for *ltgA* has not yet been identified; therefore, *ltgA* was introduced in the chromosome of meningococcus and expressed under the control of a non-native promoter. Since $\Delta ltgA^{ltgA\Delta30}$ exhibited reduced growth and this could be attributed to bacterial lysis or defects in cell division or cell separation, we examined all four strains using fluorescent microscopy (labeling the membrane and intracellular DNA), and scanning electron microscopy (SEM). Despite the reduced growth of strain $\Delta ltgA^{ltgA\Delta30}$, there was no physical evidence suggesting bacterial lysis. However, intriguingly in the $\Delta ltgA^{ltgA\Delta30}$ strain, we observedstrong defects in cell separation and cell division, and the appearance of membrane stained extracellular material that was notably absent in the other three strains (*Figure 3* (*right panel*), *Figure 3—figure supplement 1*). Additionally, SEM revealed large blebs on the surface of some of the unseparated/undivided bacteria in the $\Delta ltgA^{ltgA\Delta30}$ strain that were not observed in the other strains. A rather striking phenomenon is that the bacteria with blebs all had smooth surfaces that deviated from the normal rough surface appearance of *N. meningitidis* in the other strains (*Figure 3* (*left panel*), *Figure 3—figure supplement 1*). We also observed ghost cells; however, this phenomenon was not as pervasive as other abnormalities (*Figure 3* (*left panel*)). Interestingly, although the levels of LtgA or LtgA$^{\Delta30}$ expressed from an ectopic locus in *N.*

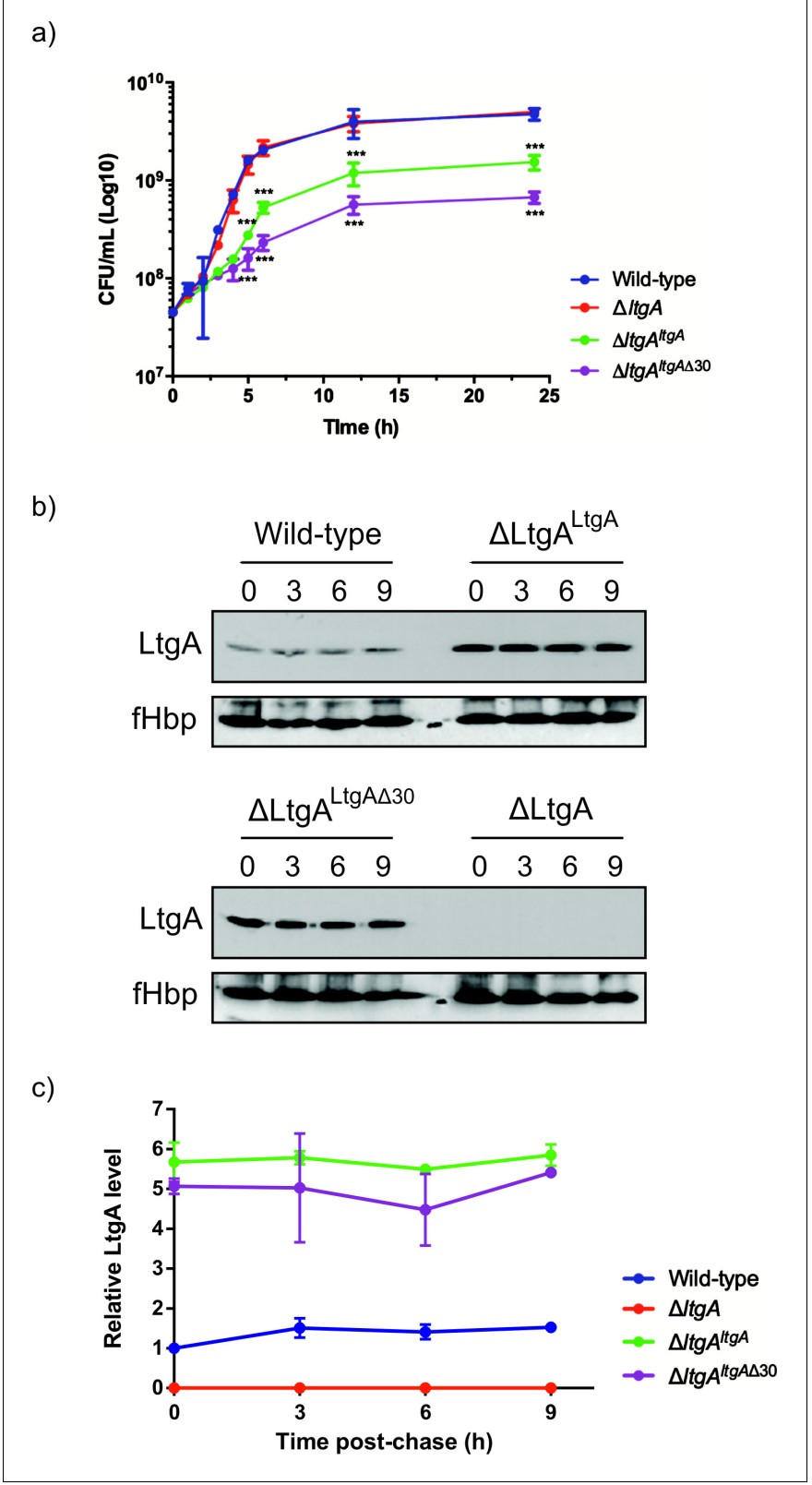

**Figure 2.** The LtgA helix 30 mutant leads to a growth defect and is stable. (a) Growth kinetics of *N. meningitidis* wild-type, Δ*ltgA*, Δ*ltgA*$^{ltgA}$ and Δ*ltgA*$^{ltgAΔ30}$ strains. Data represent three independent experiments. (b) Exponentially grown bacteria were treated with chloramphenicol (2 µg/ml) to block protein synthesis and survey the stability of LtgA for the indicated periods of time (in hours). Immunoblots were performed probing with anti-

*Figure 2 continued on next page*

*Figure 2 continued*

LtgA antibody. The expression of the outer membrane protein fHbp was used as an loading control. (c) The levels of LtgA over the time were analyzed and plotted as a stability curve by quantifying the band intensities using ImageJ software. For each strain, the LtgA intensity at time zero is referred to as 100%, while the simultaneously fHbp was used for loading control.

The online version of this article includes the following source data for figure 2:

**Source data 1.** *Figure 2a* – Analysis associated with growth kinetics of N. meningitidis wild-type, ΔltgA, ΔltgA $^{ltgA}$ and ΔltgA $^{ltgAΔ30}$ strains.

**Source data 2.** *Figure 2c* – The stability of LtgA over time.

---

*meningitidis* were higher in comparison to the natively expressed protein, severe morphological defects were only observed in the $\Delta ltgA^{ltgA\Delta30}$ strain.

## LtgA is involved in maintaining the structural composition of the peptidoglycan

We examined the PG profiles of wild-type *N. meningitidis*, $\Delta ltgA$, $\Delta ltgA^{ltgA}$ and $\Delta ltgA^{ltgA\Delta30}$, to explore whether the integrity of the PG composition of $\Delta ltgA^{ltgA\Delta30}$ strain was maintained. No notable differences were observed among the wild-type, $\Delta ltgA$ and $\Delta ltgA^{ltgA}$ strains (*Figure 4*, *Figure 4—figure supplement 1*, Supplementary 1). However, the PG of the $\Delta ltgA^{ltgA\Delta30}$ strain was found to be markedly hyperacetylated when compared to that of the other strains, with a 102% increase in the amount of acetylated GlcNAc-anhMurNAc-tetrapeptide (GM*4), a 39% increase in acetylated GlcNAc-anhMurNAc-tetrapeptide crosslinked with GlcNAc-MurNAc-tetrapeptide (GM*4-GM4), and a 46% increase in doubly acetylated di-GlcNAc-anhMurNAc-tetrapeptide (GM*4 GM*4) (*Figure 4*, *Figure 4b*). A 22% increase in the amount of GlcNAc-MurNAc tetrapeptide (GM4) was also observed, while the amounts of GlcNAc-MurNAc tripeptide (GM3) and GlcNAc-MurNAc pentapeptide (GM5) decreased by 33% (*Figure 4*, *Figure 4b*). Overall, there was a marked increase in the amounts of acetylated PG monomers and dimers.

The PG de-*O*-acetylase (Ape1) is the enzyme responsible for removing the *O*-acetyl group from the C6-hydroxyl position of the glycan strand of the *O*-acetylated PG and ensures the continued metabolism of the PG by LTs (LtgA, LtgD or LtgE and others) (*Weadge et al., 2005*; *Weadge and Clarke, 2006*; *Pfeffer and Clarke, 2012*; *Veyrier et al., 2013*). Since the PG of the $\Delta ltgA^{ltgA\Delta30}$ strain was hyperacetylated, the expression of Ape1 was assessed in all four strains (*N. meningitidis*, $\Delta ltgA$, $\Delta ltgA^{ltgA}$, $\Delta ltgA^{ltgA\Delta30}$) (*Figure 4—figure supplement 2*; *Figure 4—figure supplement 1*). Ape1 was comparably expressed in all four strains (*Figure 4—figure supplement 2*; *Figure 4—figure supplement 1*).

## The impact of protein complexes on peptidoglycan O-acetylation

Hyperacetylation of the PG was the most striking phenotype of the $\Delta ltgA^{ltgA\Delta30}$ strain. We therefore explored whether: 1) LtgA and Ape1 form a PG degrading complex, or 2) the normal function of Ape1 depends on LtgA, or 3) Ape1 and LtgA work in concert enzymatically to de-*O*-acetylate the PG. To accomplish this, we purified Ape1 and LtgA, following their heterologous expression in *E. coli* (*Figure 5a*). Each enzyme was purified individually and then combined prior to their application to size-exclusion columns (*Figure 5a*). LtgA formed an approximately 100 kDa complex with Ape1 (*Figure 5a*).

We next examined the activity of LtgA and Ape1 against acetylated PG from *N. meningitidis,* or the activity of Ape1 alone, or Ape1 combined with LtgA toward 4-nitrophenyl acetate, a previously characterized substrate of Ape1 from *N. gonorrhoeae* that is not a substrate for LtgA (*Pfeffer et al., 2013*; *Weadge and Clarke, 2007*). In the presence of equimolar (1.2 μM) amounts of Ape1, LtgA degrades the PG more efficiently (*Figure 5b*). This result is consistent with previous studies that suggest *O*-acetylation blocks the function of LTs and lysozyme (*Weadge et al., 2005*; *Weadge and Clarke, 2006*; *Pfeffer and Clarke, 2012*; *Veyrier et al., 2013*). Surprisingly, in the absence of a common substrate and utilizing equimolar amounts (12 nM) of LtgA and Ape1, LtgA enhances the activity of Ape1 (*Figure 5c*). The reaction remained well within the linear range for 60 min when both enzymes were present, which was in stark contrast to Ape1, that showed less activity over the

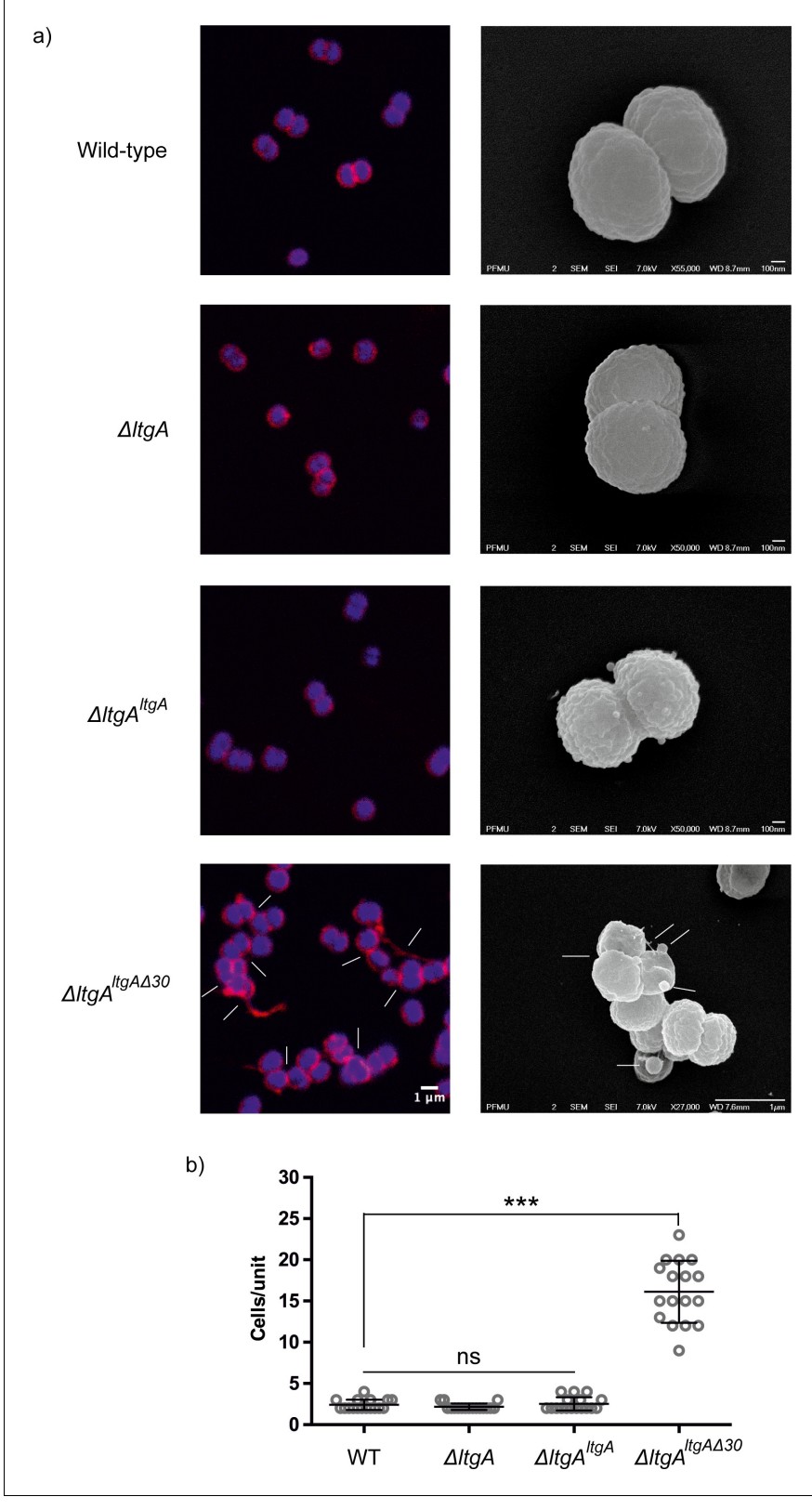

**Figure 3.** The LtgA helix 30 mutant shows morphological abnormalities. (a) Morphological differences between the wild type and strains expressing the mutant lytic transglycosylases were determined by fluorescence microscopy (*left panel*) and scanning electron microscopy (SEM) (right*panel*). White arrows in the images of ΔltgA$^{ltgAΔ30}$ strain (*left panel*) point to cells defective in division and separation, as well as extracellular material. White arrows in the right panel points to, irregular cell surfaces, high-molecular-weight blebs (not observed in other strains), asymmetrical diplococci, and ghost cells

*Figure 3 continued on next page*

*Figure 3 continued*

(see *Figure 3—figure supplement 1* for other images detailing additional morphological abnormalities). (**b**) Quantification of the confocal microscopy data. The different fields were manually counted to evaluate the number of cells per unit. Each unit is defined as an isolated cluster of cells that it is not in contact with other cells. Whenever in contact two cells were defined as apart of the same unit.

The online version of this article includes the following figure supplement(s) for figure 3:

**Figure supplement 1.** Morphological abnormalities of LtgA alpha helix 30 mutant.

**Figure supplement 2.** Morphological abnormalities of the Δape1 strain.

time course of 60 min. These data demonstrate that the enzymatic activities of LtgA and Ape1 are enhanced reciprocally when functioning together in a complex (*Figure 5c*). It also appears that LtgA stabilizes and enhance the activity of Ape1. Synergistic interaction between Ape1 and LtgA could reflect their coordinated function in PG structural regulation in vivo.

Interestingly, LtgA, along with other enzymes such as PBP1a, and LtgE are co-conserved in all the proteobacteria that were surveyed (*Figure 5—figure supplement 1*). Meanwhile, Ape1 is exclusively co-conserved in *Neisseria*, *Kingella*, *Snodgrassella*, *Morococcus*, *Azovibrio*, and one isolate of *Burkholderia ubortensis*, suggesting Ape1 in contrast to LtgA and others was potentially acquired later by lateral gene transfer (*Figure 5—figure supplement 1*).

## The source of helix 30 morphological defects

The activity of Ape one and LtgA appears to be synergistic. Since a ΔltgA gave no noticeable phenotype, *Neisseria meningitidis* strains harboring a catalytically defective mutation of LtgA (E481A), or a Δape1 strain of *N. meningitidis* were examined for morphological aberrations. The *ltgA* (E481A) strains showed no morphological abnormalities when compared to their parental strains (*Figure 3—figure supplement 2*). However, while the Δape1 strain showed no significant defects in cell division or cell separation, cell shape abnormalities and lysed bacteria were clearly evident (*Figure 3—figure supplement 2*). Additionally, in our previous study, we noted that diploid cells of the Δape1 strain were larger compared to the wild-type strain (*Veyrier et al., 2013*). Altogether these data suggest that Ape 1 is an important cell shape determinant.

## Role of alpha helix 30 in the virulence of *N. meningitidis*

In *N. meningitidis* and *N. gonorrhoeae,* the activity of LtgA and other LTs leads to a release of cytotoxic PG fragments, which are detected by the host and induce a Nod1-dependent inflammatory response (*Cloud and Dillard, 2002*; *Cloud and Dillard, 2004*; *Girardin et al., 2003*; *Schaub et al., 2016*). Since the alpha helix 30-deleted strain of LtgA compromised the PG composition the functional role of the alpha helix 30 was explored in vivo in a mouse infection model. For this purpose, we used transgenic mice expressing human transferrin as an experimental model that allows meningococcal growth by providing a human iron source during infection. The four *N. meningitidis* strains (wild-type, ΔltgA$^{ltgA}$, ΔltgA$^{ltgAΔ30}$ and ΔltgA) were used to infect the mice by intraperitoneal injection. Two hours after infection, the four strains induced similar levels of bacteremia (*Figure 6a*), suggesting that the strains were not defective in their ability to reach the bloodstream. The ΔltgA strain appears to be cleared more slowly. However, the ΔltgA$^{ltgAΔ30}$ strain was cleared from the blood at a significantly faster rate than the other strains, exhibiting a 2-log difference in colony-forming units (CFUs) at the 6 hr time point compared to the wild-type strain (*Figure 6a*). These results were also consistent with those at the cytokine production level, as the ΔltgA$^{ltgAΔ30}$ strain exhibited significantly decreased levels of IL-6 and KC (the functional murine homolog of human IL-8) 6 hr after infection, while all the isolates exhibited similar levels 2 hr after infection (*Figure 6b*). Overall, the ΔltgA$^{ltgAΔ30}$ strain displays impaired fitness in the host, suggesting LtgA alpha helix 30 plays a key role in bacterial virulence.

## Discussion

Antibiotic resistance is recognized as an urgent global public health threat. One potential solution is to develop new treatment strategies that impact multiple cellular targets and consequently, circumvent the rise of antibiotic resistance. The bacterial cell wall is assembled by a number of enzymes,

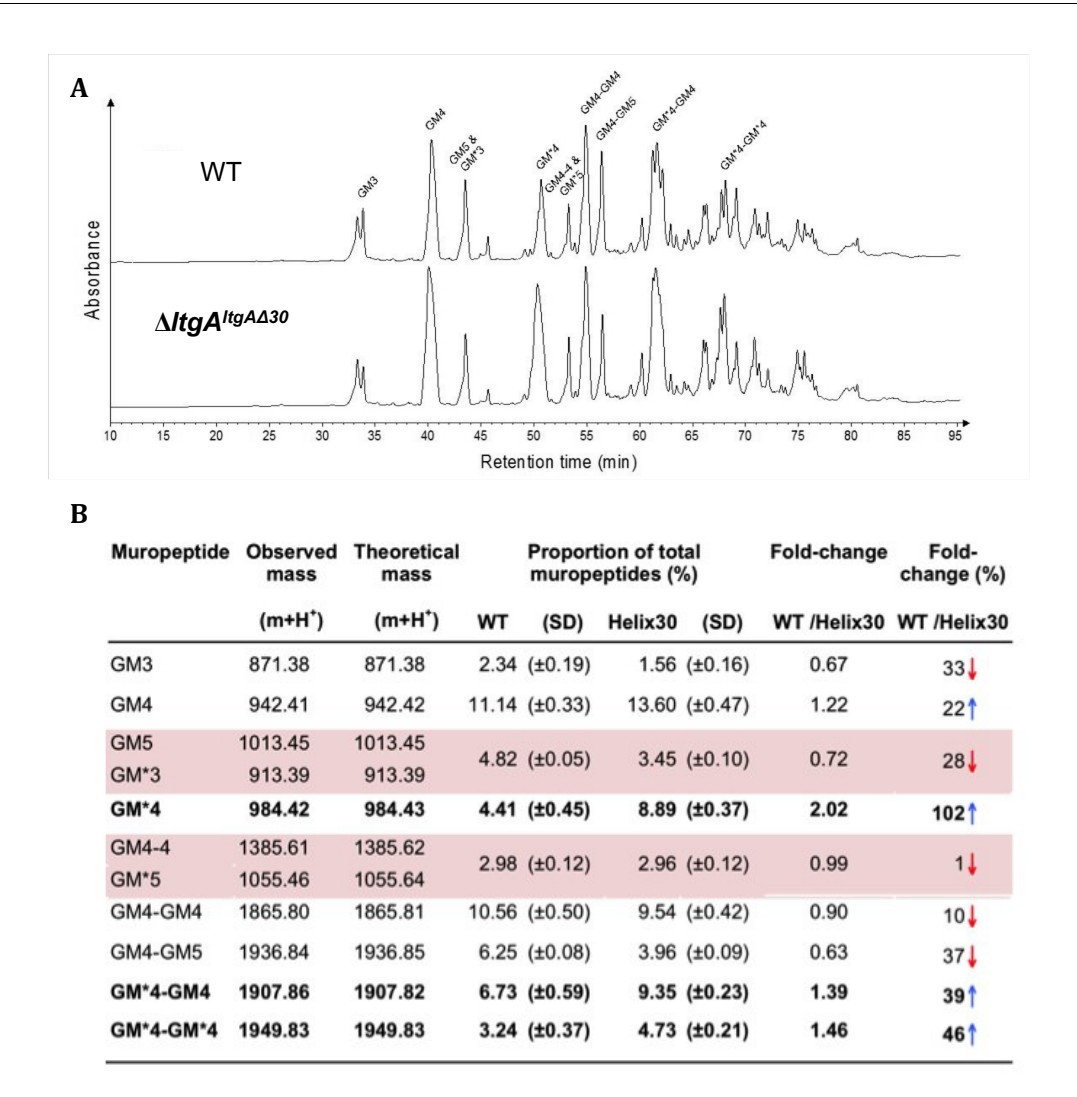

**Figure 4.** Muropeptide composition of PG isolated from wild-type and Δ*ltgA*^*ltgAΔ30*^. (a) The chromatogram represents purified PG digested by the muramidse, mutanolysin and the resulting muropeptides were reduced, and analyzed by LC/MS. The results were reproducible over four biological replicates. Peak identifications correspond to (b). Quantitation and analysis of muropeptides identified by mass spectrometry. * indicates O-acetylated MurNAc. Acetylated GM*4 is highlighted in bold. Multiple muropeptides coeluted as a single peak are shaded in pink. Red arrows indicate a decrease and blue arrows an increase in muropeptide abundance. The table displays the observed and theoretical masses and the proportion of total muropeptides.

The online version of this article includes the following source data and figure supplement(s) for figure 4:

**Source data 1.** Table associated with the muropeptide composition of PG isolated from wild-type, Δ*ltgA*, Δ*ltgAltgA*.
**Source data 2.** Raw data associated with themuropeptide composition of PG isolated from wild-type and Δ*ltgAltgAΔ30*.
**Figure supplement 1.** Muropeptide composition of PG isolated from wild-type, Δ*ltgA*, Δ*ltgA*^*ltgA*^.
**Figure supplement 2.** The expression of Ape1.

some of which are broadly categorized as PG polymerases, PG-modifying enzymes and PG hydrolases. The primary targets of β-lactams, clinically the most utilized antibiotics, are PBPs that are known to polymerize PG. The development of a combinatorial or single therapy that interferes with multiple cellular function of the PG machinery could define a new era in antibiotic development. This would be particularly relevant for *N. gonorrhoeae* infections, as no vaccines against this species are currently available, and highly resistant strains are on the rise. Indeed, a *N. gonorrhoeae* 'superbug' has already been identified that does not respond to the usual treatment with β-lactams such as ceftriaxone (*Suay-García and Pérez-Gracia, 2017*).

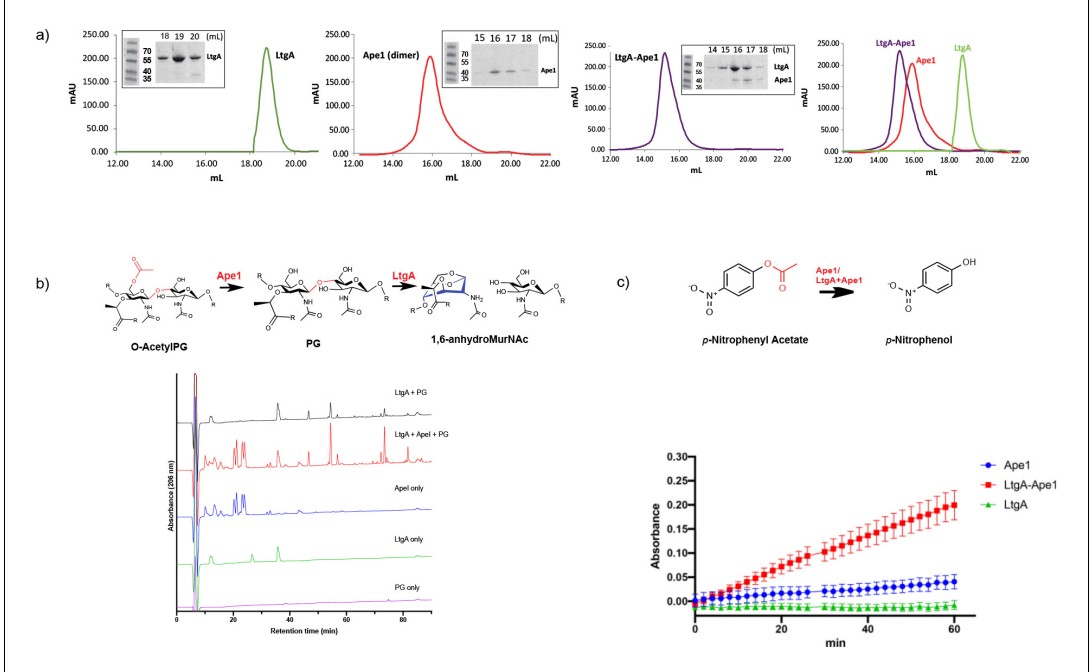

**Figure 5.** LtgA stimulates and stabilizes the enzymatic activity of Ape1. (**a**) Chromatogram of the size exclusion analysis of purified (*panel 1*) LtgA, (*panel 2*)Ape1, (*panel 3*) Ape1-LtgA 105 kDa. Overlay of the chromatograms corresponding to purified LtgA, Ape1, and LtgA-Ape1 protein complex (*panel 4*). Each insert represents SDS-PAGE analysis of peak fractions containing proteins. Lanes are labeled with corresponding volumes. (**b**) Ape1 stimulates the activity of LtgA toward the acetylated PG of *N. meningitidis*. (**c**) LtgA stimulates and stabilizes Ape1 activity toward *p*-nitrophenol acetate. Ape1 utilizes *p*-nitrophenol acetate as a substratewhile LtgA does not. Error bars show the standard deviation of triplicate.

The online version of this article includes the following source data and figure supplement(s) for figure 5:

**Source data 1.** *Figure 5b* – LtgA stimulation assay.
**Source data 2.** *Figure 5c* – Ape1 stimulation assay.
**Figure supplement 1.** Phylogenetic tree showing the the co-conservation of Ape1.

In this study, we identified a variant of LtgA with a disordered active site alpha helix 30, which is important for PG binding and for the catalytic mechanism of LtgA (*Figure 1b*, *Video 1*). LTs are highly redundant enzymes, and when individual or multiple LTs are deleted, bacteria are known to proliferate normally because others LTs compensate for the loss of activity and/or function (*Cloud and Dillard, 2002*; *Lee et al., 2013*). Interestingly, when 6 LTs were deleted from *E. coli*, only a mild chaining phenotype was observed (*Heidrich et al., 2002*); however, a mutant of *ltgC* with a 33 bp deletion at the 5′ end from *Neisseria* sp., showed defects in growth and daughter cell separation (*Cloud and Dillard, 2004*). In our Δ*ltgA*$^{ltgAΔ30}$ strain, we observed significant defects in growth, cell division, cell separation, cell membrane irregularities, and fibrous and membranous extra cellular material, which were noticeably absent in the wild type, Δ*ltgA*, and the Δ*ltgA*$^{ltgA}$ strains (*Figures 2* and *3*, *Figure 3—figure supplement 1*). LtgA has been shown to localize to the septum (*Schaub et al., 2016*), but no phenotype associated with cell division or cell separation was previously reported. However, the expression of an LT with impaired function (LtgA$^{Δ30}$) results in the perturbation of various processes that are essential for bacterial proliferation.

Having observed that the Δ*ltgA*$^{ltgAΔ30}$ strain, but not the Δ*ltgA* strain, was defective in growth, cell separation, and cell division, we then analyzed the PG profiles of wild-type *N. meningitidis*, Δ*ltgA*, Δ*ltgA*$^{ltgA}$ and Δ*ltgA*$^{ltgAΔ30}$, and observed that Δ*ltgA*$^{ltgAΔ30}$ strain was hyperacetylated and there was an increase in PG monomers (*Figure 4*, *Figure 4b*). Since the hyperacetylation of the PG was the most striking phenotype, we analyzed the functional relationship between LtgA and Ape1. The interaction between LtgA and Ape1 was not unexpected because Ape1, a PG-de-*O*-acetylase, removes the *O*-acetyl group from the C6-hydroxyl position of the glycan strand of *O*-acetylated PG and ensures the continued metabolism of the PG by LTs (LtgA, LtgD or LtgE and others) (*Weadge et al., 2005*; *Weadge and Clarke, 2006*; *Pfeffer and Clarke, 2012*; *Veyrier et al., 2013*).

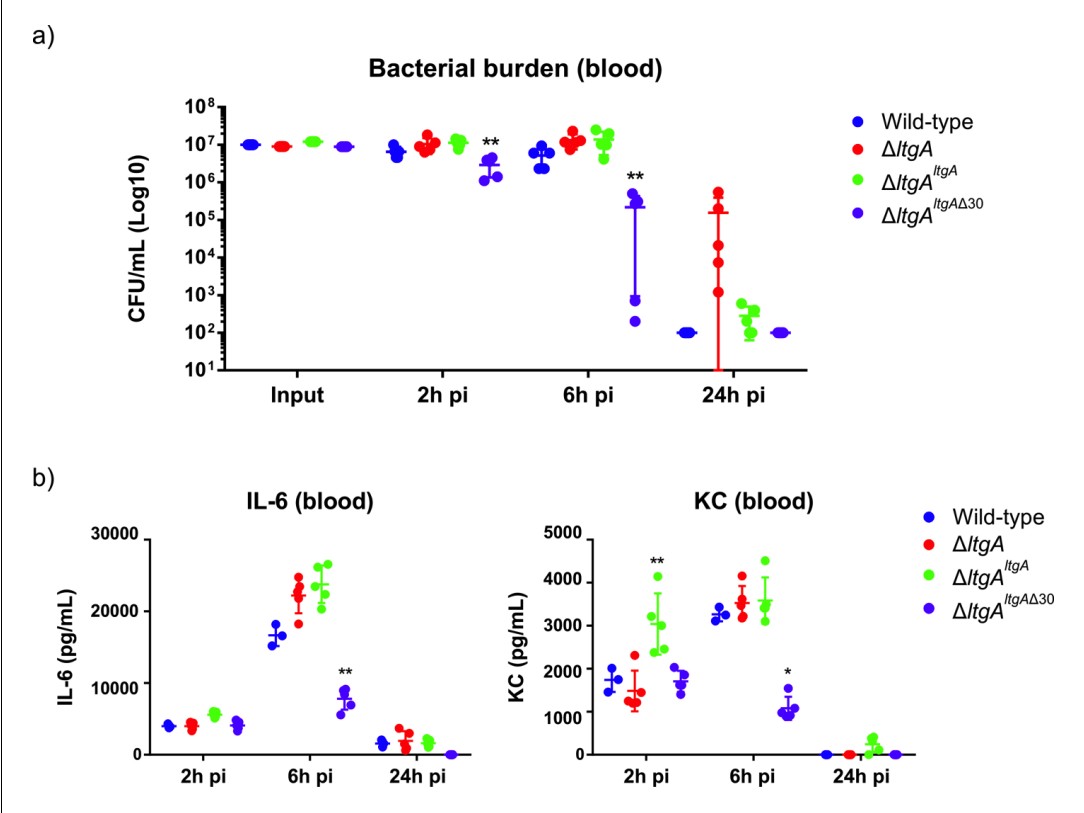

**Figure 6.** Helix 30 of LtgA plays a role in *N.meningitidis* host adaptation and virulence. *N.meningitidis* wild-type, Δ*ltgA*, Δ*ltgA*$^{ltgA}$ and Δ*ltgA*$^{ltgAΔ30}$ were administered to transgenic mice expressing human transferrin via an intraperitoneal route. (a) Bacterial burden was determined by enumeration of CFUs in blood 2, 6 and 24 hr pi. These data show that the strain complemented with a deletion in helix 30 is cleared faster than the other strains. (b) Pro-inflammatory cytokine (IL-6) and chemokine (KC) profile in blood of infected mice was evaluated 2, 6 and 24 hr post-infection by ELISA. The Δ*ltgA*$^{ltgAΔ30}$ strain induced the production of lower levels of inflammatory mediators production upon infection compared to the other strains. Data represent three independent experiments with *n* = 5. Statistical analysis was done by Kruskal-Wallis non-parametric comparison against the complemented strain with a p-value<0.01.

The online version of this article includes the following source data for figure 6:

**Source data 1.** *Figure 6a* – Quantification of bacterial burden.
**Source data 2.** *Figure 6b* – Raw files associated with pro-inflammatory cytokines.

Based on the hyperacetylation of the PG, we hypothesized that a functional LtgA is needed to stablize the activity of Ape1 and together they work in concert to ensure the proper metabolism of the PG (*Figure 5b–c*). To test this, we used 4-nitrophenyl acetate, a known substrate of Ape1 but not LtgA. Surprisingly, LtgA enhances the activity and function of Ape1 well past the normal range (1 hr) of most in vitro enzymatic reactions, giving clear evidence that LtgA could orchestrate the activity, and function of Ape1 (*Figure 5b–c*). Since Ape1 and LtgA activity appears to be synergistic, we examined whether the abberant phenotype of the Δ*ltgA*$^{ltgAΔ30}$ strain could be related to the malfunction of Ape1. Similar to the Δ*ltgA*$^{ltgAΔ30}$ strain, the Δ*ape1 strain* showed clear cell membrane irregularities; however, there were no significant defects in cell division or separation (*Figure 3—figure supplement 2*). The observed cell membrane irregularities of the Δ*ltgA*$^{ltgAΔ30}$ strain (*Figure 3*) could be due to the malfunction of Ape1, and defects in cell division or separation could be linked to a dysfunctional LtgA. A recent study showed that in *Vibrio cholerae*, LTs RlpA and MltC similar to LtgA both localize to the septum and contribute to daughter cell separation suggesting that during septal PG synthesis glycan strands are formed between daughter cells (*Weaver et al., 2019*).

Our study revealed an intimate relationship between LtgA and Ape1, ie. LtgA enhances the activity of Ape1 and an impaired LtgA results in a dysfunctional Ape1, and appears to poison the cell wall machinery with devastating effects toward the survival of *Neisseria in the host*.

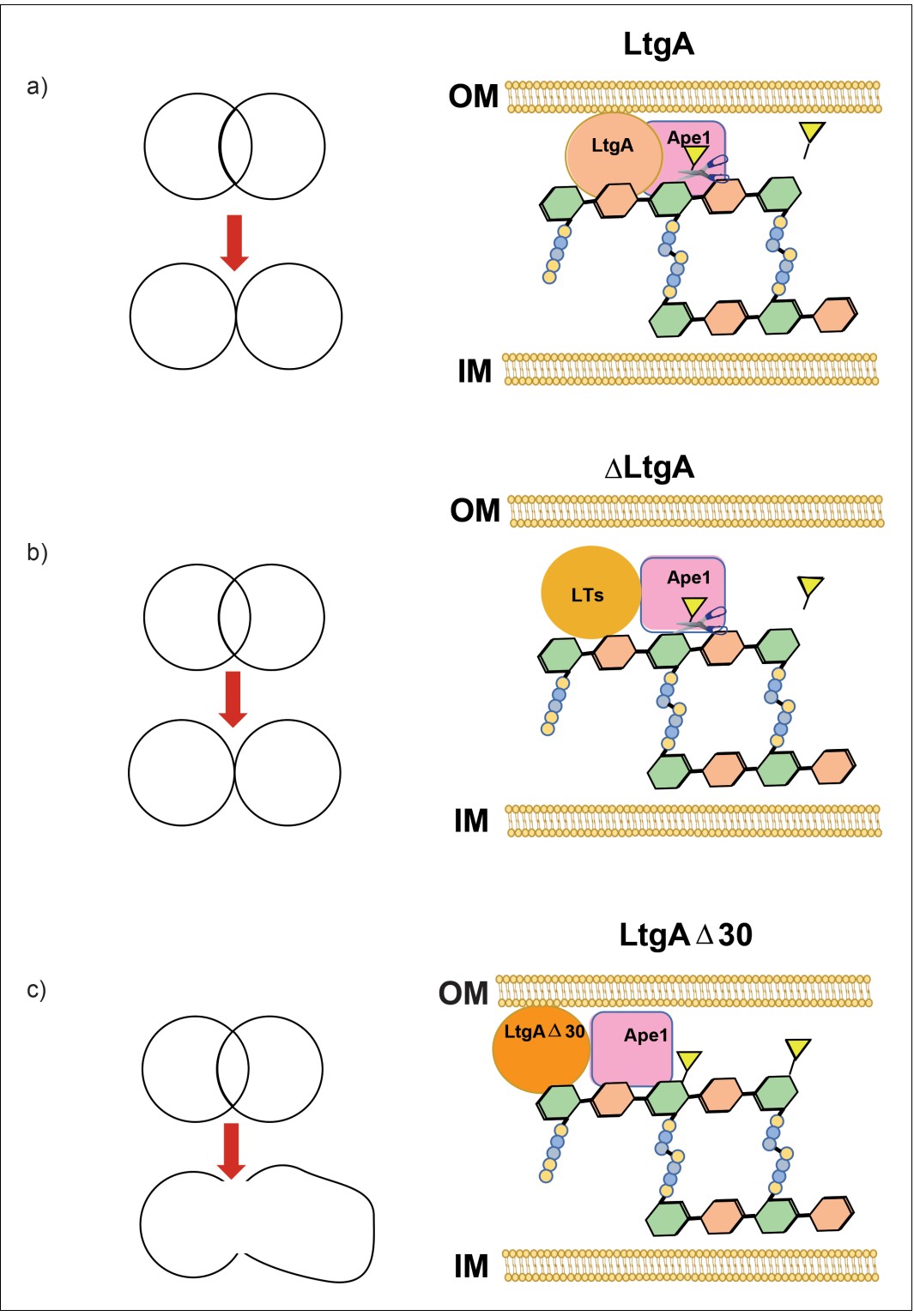

**Figure 7.** Peptidoglycan degrading complexes can modulate enzymatic function. (**a**) The removal of the *O*-acetyl group by Ape1 allows LtgA to efficiently metabolize the PG. There was normal cell growth, division and separation (yellow triangle represent acetyl groups, glycan strand are colored in green and orange, and stem peptides are represented in blue and yellow circles). (**b**) Removal of LtgA does not affect the function of Ape1. Normal cell growth, division and or separation was observed. (**c**) Deletion of alpha helix 30 of LtgA affects the function of Ape1, disrupts, growth, cell division and cell separation and bacterial survival in the host.

Finally, to understand what role a defective LtgA that interferes with the normal function of the PG machinery plays in the pathogenesis of *N. meningitidis*, we used a mouse infection model and showed $\Delta ltgA^{ltgA\Delta30}$ strain of *N. meningitidis* was cleared from the blood at a significantly faster rate than the wild-type, $\Delta ltgA^{ltgA}$ or $\Delta ltgA$ strains. Consistent with the virulence phenotype, and in comparison to the other three strains, the $\Delta ltgA^{ltgA\Delta30}$ strain in mice resulted in significantly decreased levels of IL-6 and KC, and decreased bacterial load in the blood at 6 hr post-infection (*Figure 6a–b*), indicating a loss of fitness of the helix-30 deleted mutant strain in the host.

One possible explanation for the differences in bacterial load after 6 hr in the helix-30 deleted strain is bacterial clearance mediated by the complement system. It is well accepted that *Neisseria meningitidis* is eliminated from the blood through complement lysis (*Schneider et al., 2007*). Generally, individuals with complement lysis difficiencies are at a higher risk for invasive meningococcal disease. Additionally, the helix deleted strain is hyperacetyaled and it is known that modification of the PG makes the cell wall more susceptible to complement-mediated lysis (*Zarantonelli et al., 2013*; *Rosain et al., 2017*), and PG modification is also associated with a decreased inflammatory response (*Taguchi et al., 2019*).

In summary, Ape1's activity is enhanced by LtgA. An impaired LtgA disrupts the function of Ape1, the normal course of bacterial cell division or cell separation (*Figure 7a-c*), paving the way for the design of inhibitors or antibiotics that target LTs.

## Conclusion

We devised a multidisciplinary approach using structural biology to show that it is possible to target a 'hot spot' on an LT in order to affect bacterial growth, cell division, and cell membrane integrity (*Figure 7a-c*), which resulted in lethal consequences for the bacteria during host infection. Additionally, as we discovered with Ape1, LTs can regulate the function and activity of their protein binding partners, revealing an additional role for LTs in the bacterium . This study shows the ripple effects of disrupting LtgA PG binding capabilities and sets the stage for future development of a class of antibiotics that may act by a dual action in vivo. A small molecule binding to alpha helix 30 could interfere with growth and simultaneously promote bacterial clearance, mimicking the enhanced clearance of the $ltgA^{ltgA\Delta30}$ mutant in a murine infection model.

# Materials and methods

## Key resources table

| Reagent type (species) or resource | Designation | Source or reference | Identifiers | Additional information |
|---|---|---|---|---|
| Gene (include species here) | ltgA (*Neiserria meningitidis MC58/*8013) | This paper | | |
| Strain, strain background (include species and sex here) | *ltgA^{ltgA\Delta30} Neiserria meningitidis MC58/*8013) | This paper | | |
| Strain, strain background (include species and sex here) | *ltgA^{ltgA}* (*Neiserria meningitidis MC5/*8013) | This paper | | |
| Strain, strain background (include species and sex here) | *^{ΔltgA}* (*Neiserria meningitidis MC58/*8013 | This paper | | |
| Strain, strain background (include species and sex here) | *^{Δape1}Neiserria meningitidis MC58* | doi: 10.1111/mmi.12153 | | |

*Continued on next page*

*Continued*

| Reagent type (species) or resource | Designation | Source or reference | Identifiers | Additional information |
|---|---|---|---|---|
| Strain, strain background (include species and sex here) | [ltgA(E481A)]*Neiserria meningitidis* 8013 | This paper | | |
| Antibody | LtgA Guinea pig polyclonal Antibody | This paper | | 1:250 |
| Antibody | Ape1 mouse polyclonal Antibody | This paper | | 1:250 |
| Recombinant DNA reagent | pGEX-4T1-LtgA | doi: 10.3390/antibiotics6010008 | | |
| Recombinant DNA reagent | pGEX-4T1-Ape1 | doi: 10.1111/mmi.12153 | | |
| Software, algorithm | GraphPad Prism | | RRID:SCR 002798 | https://www.graphpad.com/scientific-software/prism/ |

## Protein expression and purification

All constructs were created using standard molecular biological techniques. All constructs used for protein expression and purification in this study were GST fusions expressed from pGEX-4T1 (GE Life Sciences). The native proteins without signal peptides were expressed in BL21(DE3) Gold competent cells (Novagen). The gene encoding the LtgA deletion mutant lacking the alpha helix [503](ATAREIAGKIGMD)[513] was chemically synthesized by ProteoGenix. The synthesized *ltgA* alpha helix deletion gene was cloned into a GST-fusion pGEX-4T1 (GE Life Sciences) plasmid as described above. The expression of all constructs was induced with 0.6 mM IPTG at an optical density at 600 nm (OD600) of 0.7–0.8 and harvested after 4 hr of induction at 18˚C. After glutathione-affinity chromatography and thrombin cleavage, proteins were purified to homogeneity by size-exclusion chromatography (Superdex-200, GE) in 50 mM HEPES (pH 7.4), 150 mM NaCl, and 1 mM BME. After gel filtration, the proteins were immediately used for crystallization or flash frozen in liquid nitrogen and stored at −80˚C.

## X-ray crystallography

Crystallization screening was carried out by the sitting-drop vapor-diffusion method with a Mosquito (TTP Labtech) automated crystallization system. All crystals were grown at 18˚C using the hanging-drop vapor-diffusion method. Crystals of 15–20 mg/ml LtgA were grown at 18˚C and appeared within 2–3 days. LtgA was crystallized in a 1:1 (v/v) ratio against a well solution of 33% (w/v) PEG 6000 and 100 mM HEPES, pH 7.5. Crystals were rectangular in shape and grew to approximately 200–300 μm in length.

The data set was collected at the Soleil Synchrotron (Beamline Proxima-1) (*Supplementary file 2*). Phasing by molecular replacement was performed using Phenix (*Adams et al., 2010*). Building was performed using Coot (*Emsley and Cowtan, 2004*), and restrained refinement was carried out using a combination of Phenix and the ccp4 software suite (*Adams et al., 2010*; *Collaborative Computational Project, Number 4, 1994*). MolProbity was used during building and refinement for iterative structure improvements (*Davis et al., 2004*).

All structural figures were generated with PyMOL (PyMOL Molecular Graphics System, version 1.5, Schrödinger, LLC). The crystallographic parameters, data statistics, and refinement statistics are shown in *Supplementary file 2*. Modeling of unknown LTs were accomplished using Phyre 2 (*Kelley et al., 2015*). Videos of the LtgA enzymatic steps were generated in PyMOL and then assembled in photoshop and edited in iMovie.

## Protein-protein interaction studies by gel filtration

To explore the interactions of LtgA and its PG binding partners, proteins were mixed at equimolar concentrations of 10 μM, incubated on ice for 1 hr, and subjected to gel filtration chromatography on an SD200 10/300 column. Approximately 150–300 μl of each sample was applied to the column

in 50 mM HEPES (pH 7.5) and 150 mM salt. Peak fractions were then subjected to SDS-PAGE (5–15%) analysis.

## Analysis of *Neisseria* sp. peptidoglycan by reversed-phase HPLC and mass spectrometry

The peptidoglycan isolated from all four strains (wild-type, $\Delta ltgA$, $\Delta ltgA^{ltgA}$ and $\Delta ltgA^{ltgA\Delta30}$) was incubated for 16 hr in the presence of 10 µg of mutanolysin in 12.5 mM sodium phosphate buffer (pH 5.6) at 37°C (total reaction volume 150 µl). The reaction was stopped by boiling the samples for 3 min, and the supernatant containing the soluble muropeptides was collected after centrifugation at 16,000 $\times g$ for 10 min. The supernatant was analyzed by reversed-phase HPLC using a Hypersil GOLD aQ column (5 µm particle size, 150 × 4.6 mm, flow rate 0.5 ml at 52°C, Thermo Fisher Scientific) with a mobile phase of $H_2O$-0.05% trifluoroacetic acid and a 25% acetonitrile gradient over 130 min. Muropeptides of interest were collected and identified by mass spectrometry as previously described (*Williams et al., 2017*; *Williams et al., 2018*).

## Bacterial strains

Clone 12 is a derivative of strain 8013, a serogroup C *N. meningitidis* strain (*Nassif et al., 1993*), and MC58 is a serogroup B strain (*Tettelin et al., 2000*). Bacteria were grown on GCB medium (Difco) containing Kellogg's supplements (*Kellogg et al., 1963*). The *E. coli* strain DH5 (*Hanahan, 1983*) was used for plasmid preparation and subcloning. Kanamycin, ampicillin and erythromycin were used in *E. coli* at final concentrations of 50, 100 and 300 µg/ml, respectively. In *N. meningitidis*, kanamycin, ampicillin and erythromycin were used at final concentrations of 2, 20 and 100µg/ml, respectively.

## Plasmid construction

The *ltgA* gene (1851 nucleotides according to the genome sequence of the meningococcal strain MC58) was chemically synthesized with a deletion of 42 bp (14 codons) between positions 1506 (codon 502) and 1548 (codon 516) (starting at ATG) and was cloned into the vector pUC57 to generate the recombinant plasmid pUC57ltgA (ProteoGenix, Schiltigheim, France). The *ltgA* fragment was amplified using the primer pair NMF1/NMR1 from the plasmid pUC57ltgA and from the strain MC58. The two fragments were blunt-ended using the Klenow DNA polymerase fragment (BioLabs) and subcloned into the *BamH*I site of the recombinant plasmid pTE-KM (*Taha et al., 1998*). This plasmid contains the *pilE* gene of clone 12 with the Km cassette, encoding resistance to kanamycin, located immediately downstream of the *pilE* without modification of *pilE* expression. Moreover, a unique BamHI site located between the Km cassette and the downstream region at the 3' end of the *pilE* gene (*Nassif et al., 1993*; *Taha et al., 1998*) was used to subclone the two blunt-ended fragments from the plasmid pUC57ltgA and from the strain MC58 to yield the recombinant plasmids pD-$\Delta ltgA^{ltgA}$ and pD-$\Delta ltgA^{ltgA\Delta30}$, respectively.

An internal deletion in the *ltgA* gene was also constructed by removing the segment between the restriction sites BsmI (position 21) and BalI (position 1724) on pUC57ltgA, and this region was replaced with the *ermAM* cassette, encoding erythromycin resistance; the construct was checked using the primer pair ERAM1/ERAM3 (5'-gcaaacttaagagtgtgttgatag-3' and 5'-aagcttgccgtctgaatgg-gacctctttagcttcttgg-3', respectively) (*Taha et al., 1998*). The corresponding recombinant plasmid pDG15-09 was linearized at the *EcoR*I site of the pUC57 vector and used to transform the clone 12 strain of *N. meningitidis*. Transformants were selected on standard GCB medium in the presence of 2 µg/ml erythromycin. Integration by homologous recombination into the *ltgA* gene on the meningococcal chromosome was further confirmed by PCR analysis using the oligonucleotides ERAM1/ERMA3 and NMF1/NMR1. One transformant was selected for further analysis and named pD-$\Delta ltgA$.

The two recombinant plasmids pD-$\Delta ltgA^{ltgA}$ and $\Delta ltgA^{ltgA\Delta30}$ were linearized using the ScaI restriction enzyme and used to transform the strain pD-$\Delta ltgA$. Transformants were selected on standard GCB medium in the presence of 2 µg/ml erythromycin and 100 µg/ml kanamycin. Integration by homologous recombination into the *ltgA* gene on the meningococcal chromosome was further monitored by PCR analysis using the oligonucleotides pilE1, NMF1, NMR1 and NMF1/NMR1. One transformant from each transformation was selected for further analysis and named $\Delta ltgA^{ltgA}$ or $\Delta ltgA^{ltgA\Delta30}$. The strain $\Delta ltgA^{ltgA}$ has the *ltgA* gene deleted from its locus but harbors the *ltgA* gene

downstream of the *pilE* site. The Δ*ltgA*$^{ltgAΔ30}$ strain also has the *ltgA* gene deleted from its locus and contains a downstream *pilE* gene but harbors the *ltgA* gene with the region encoding the amino acid residues 501–516 deleted.

Strains 8013 expressing mutant lytic transglycosylases (E481A) were constructed by transformation with plasmid pRS91 (*Schaub et al., 2016*) containing the E481 mutation. Potential transformants were screened by PCR amplification of active site region followed by digestion with Hyp188III. Positive transformants that lacked a Hyp188III site at the active site were confirmed by sequencing.

The MC58 Δ*ape1 strain* was described in our previous study (*Veyrier et al., 2013*). Briefly, the entire *pat* operon which consists of *patA*, *patB* and *ape1* was deleted in MC58 and then this knockout mutant was complemented by the introduction of *patA* and *patB genes*.

## Fluorescent labeling and fluorescent microscopy

Bacterial cultures were centrifuged 5 min at 5000 rpm and re-suspended in PBS containing 1 μg/mL DAPI and 5 μg/ml FM4-64 FX (*N*-(3-Triethylammoniumpropyl)−4-(6-(4 (Diethylamino) Phenyl) Hexatrienyl) Pyridinium Dibromide) probe. The cells were incubated for 10 min at room temperature protected from light, centrifuged and the pellets resuspended in 4% PFA for fixation during 5 min. After fixation, the cells were washed with PBS, and a 10 μL drop of the bacterial suspension was applied onto poly-L-Lysine pre-coated cover glasses (# 1.5). Next, samples were mounted using Prolong Diamond and imaged using Leica SP5 confocal microscope, with a 63X (1.4 NA) oil-immersion objective using 405 nm and 514 nm laser lines. Fluorescence was recorded sequentially using hybrid (HyD) detectors and images processed using Fiji (*Schindelin et al., 2012*).

## Scanning electron microscopy

*Neisseria meningitidis* was prefixed in 2.5% Glutaraldehyde diluted in PHEM (Pipes, Hepes, EGTA and MgSO$_4$) buffer at pH 7. The cells were prefixed for 1 hr at room temperature, followed by two washes in PHEM buffer. The samples were applied onto the cover glass (1.5 mm) pre-coated with poly-L-Lysine. This was followed by a low speed centrifugation to ensure that the cells adhere correctly to the cover slip.

The bacteria were post-fixed using 2% osmium tetroxide in PHEM buffer for 30 to 60 min followed by washing with water three times. The specimen was dehydrated using increasing ethanol concentrations of 25% to 100% in increments of 25%. The bacteria were critically point dried using carbon dioxide, coated with gold and examined with the JEOL JSM-6700F scanning electron microscope.

## *O*-acetyl peptidoglycan esterase assay

The acetyl esterase activity assays were executed as previously described with minor modifications (*Pfeffer et al., 2013*; *Hadi et al., 2011*). Briefly, the reaction utilized 2 mM 4-nitrophenyl acetate as the substrate. The reaction was carried out at 37°C in 50 mM sodium phosphate buffer, pH 6.5 in the presence of LtgA, using equimolar amounts of LtgA and Ape1 or Ape1. The final volume of the reaction was 300 μl. The reaction was initiated with the addition of the substrate 4-nitrophenyl acetate dissolved in 5% v/v ethanol. The release of 4-nitrophenyl was monitored over the time course of an hour in 96 well microtiter plate at an absorbance of 405 nm.

## Analysis of LtgA activity

To assess the activity of LtgA, PG (200 μg) was incubated in the presence of LtgA, or equimolar amounts of LtgA and Ape1, in 12.5 mM sodium phosphate buffer pH 5.6. *Neisseria* PG was purified as previously described (*Wheeler et al., 2014*). The reaction mix was initiated by the addition of enzymes and incubated at 37°C for 5 min. Control reactions lacking PG or enzyme/inhibitor were also included. The final reaction volume was 200 μL. Reactions were performed in triplicates. The reaction was stopped by incubating the samples in a heat block at 100°C for 5 min. The soluble 1,6-anhydro-muropeptides was collected using centrifugation at 16,000 *g* for 10 min at room temperature. The supernatant was collected and analyzed by reversed-phase HPLC using a Shimadzu LC-20 system with a Hypersil GOLD aQ column (5 μm particle size, 250 × 4.6 mm, flow rate 0.5 mL/mL at 52°C; Thermo Fisher Scientific (Waltham, MA, USA). The mobile phase gradient was 50 mM sodium phosphate pH 4.3 to 75 mM sodium phosphate pH 4.9 with 15% Methanol over 135 min.

## Infection model

A previously published model for meningococcal infection in transgenic mice expressing human transferrin was used (*Szatanik et al., 2011*). Four strains were tested: clone 12 (wild-type), $\Delta ltgA^{ltgA}$, $\Delta ltgA^{ltgA\Delta30}$ and $\Delta ltgA$. Five mice per group were infected by intraperitoneal injection with 500 µl of bacterial suspension of each strain at $1 \times 10^7$ CFU/ml. Blood samples were obtained by retro-orbital bleeding after 2, 6 and 24 hr, and bacterial counts were determined by plating serial dilutions on GCB medium.

## Phylogenetic tree construction

Protein sequences were aligned using MUSCLE alignment algorithm using UPGMA clustering method in MEGAX (*Kumar et al., 2018*). Using aligned sequences, a maximum likelihood tree was constructed using a Neighbor joining construction method and a JTT protein substitution model in CLC Genomics Workbench 8.01. Robustness was estimated using 500 bootstrap replicates (values not shown in figures).

## Ethics statement

Animal work in this study was carried out at the Institut Pasteur in strict accordance with the European Union Directive 2010/63/EU (and its revision 86/609/EEC) on the protection of animals used for scientific purposes. The laboratory at the Institut Pasteur has the administrative authorization for animal experimentation (Permit Number 75–1554) and the protocol was approved by the Institut Pasteur Review Board that is part of the Regional Committee of Ethics of Animal Experiments of Paris Region (Permit Number: 99–174). All the invasive procedures were performed under anesthesia and all possible efforts were made to minimize animal suffering.

## Cytokine assay

Blood samples from infected mice were collected and stored at $-80°C$. Cytokines (IL-6 and KC) were quantified by an enzyme-linked immunosorbent assay (Quantikine; R and D Systems Europe, Abingdon, Oxon, United Kingdom).

## Growth curves and LtgA stability assay

Bacteria were grown overnight in GC broth with Kellogg's supplements at 37°C and 5% $CO_2$. Fresh medium was inoculated at an OD600 of 0.05, and growth was measured spectrophotometrically at 1 hr intervals over a period of 24 hr at 37°C and 5% $CO_2$. When indicated, 2 µg/ml chloramphenicol was added when the $OD_{600}$ reached 0.6, and incubation was continued at 37°C and 5% $CO_2$. At different incubation time points, aliquots (3 ml) from each culture were sampled, and the bacteria were collected by centrifugation, lysed by boiling in SDS-containing sample buffer, and analyzed for the presence of LtgA by western blotting using anti-LtgA antibodies. The expression of the outer membrane factor H binding protein (Fhbp) was used as an internal control.

## PG binding assay

The binding of the different LtgA proteins to PG was carried out by incubating 100 µg of PG and 10 µg of enzymes suspended in 150 µl of Tris buffer pH 7.5 (10 mM Tris, 10 mM $MgCl_2$ and 50 mM NaCl). After 30 min of rocking at room temperature, 50 µl of the sample was set aside for analysis before centrifugation for 10 min at 20,000 x*g*. The supernatant was discarded, and the insoluble fraction was washed three times. The remaining pellet was boiled for 10 min. Five microliters of the input or unbound and bound fractions was loaded on an SDS-PAGE gel and analyzed by western blotting.

## Accession numbers

Coordinates and structural data have been submitted to the Protein Data Bank under the accession code 6H5F.

## Acknowledgements

We acknowledge the synchrotron beamline staff (PROXIMA-1 at SOLEIL and X06DA at SLS) for their assistance. We are extremely grateful to Frederick Saul, Patrick Weber and Marco Bellinzoni for their constant helpful guidance, advice and assistance. We thank Dr. Antoine Forget for the advice on and help with figure presentations. AHW was supported by an EMBO long-term fellowship (ALTF 732–2010) and an Institut Carnot-Pasteur Maladies Infectious fellowship. This work was supported by an ERC starting grant (PGN from SHAPE to VIR 202283) and a Fondation pour la recherche médicale (FRM) grant Programme d'Urgence (DBF20160635726) to IGB. This study received funding from the French Government's Investissement d'Avenir program, Laboratoire d'Excellence 'Integrative Biology of Emerging Infectious Diseases' (grant no. ANR- 10-LABX-62-IBEID). IS was supported by the Institut Carnot Pasteur Microbes and Santé given to the Pasteur-Paris University PhD program and the 'Fin de these de science' number FDT201805005258 granted by "Fondation pour la recherche médicale (FRM). The UtechS Photonic BioImaging (Imagopole), C2RT, Institut Pasteur was supported by the French National Research Agency (France BioImaging; ANR-10–INSB–04; Investments for the Future).

## Additional information

### Funding

| Funder | Grant reference number | Author |
| --- | --- | --- |
| European Molecular Biology Organization | ALTF 732-2010 | Allison Hillary Williams |
| European Research Council | PGN from SHAPE to VIR 202283 | Ivo Gomperts Boneca |
| Fondation pour la Recherche Médicale | DBF20160635726 | Ivo Gomperts Boneca |
| Institut Carnot-Pasteur | Maladies Infectious fellowship | Allison Hillary Williams |
| Institut Carnot Pasteur Microbes and Sante | | Ignacio Santecchia |
| Fondation pour la Recherche Médicale | FDT201805005258 | Ignacio Santecchia |

The funders had no role in study design, data collection and interpretation, or the decision to submit the work for publication.

### Author contributions

Allison Hillary Williams, Conceptualization, Data curation, Formal analysis, Supervision, Investigation, Methodology, Project administration; Richard Wheeler, Formal analysis, Validation, Investigation, Methodology; Ala-Eddine Deghmane, Ignacio Santecchia, Validation, Investigation, Methodology; Ryan E Schaub, Resources, Investigation, Made the strain ltgA (E481A) that was an important control, Gave input on the final manuscript; Samia Hicham, Ahmed Haouz, William P Robins, Muhamed-Kheir Taha, Resources, Investigation, Methodology; Maryse Moya Nilges, Validation, Investigation; Christian Malosse, Resources, Formal analysis, Investigation; Julia Chamot-Rooke, Resources, Investigation; Joseph P Dillard, Resources, Methodology, The lab of Dr. Dillard created and provided the strain of ltga (E481A) which was an important control we needed after revision; Ivo Gomperts Boneca, Conceptualization, Formal analysis, Supervision, Funding acquisition, Project administration

### Author ORCIDs

Allison Hillary Williams (iD) https://orcid.org/0000-0002-0726-141X

### Ethics

Animal experimentation: Animal work in this study was carried out at the Institut Pasteur in strict accordance with the European Union Directive 2010/63/EU (and its revision 86/609/EEC) on the

protection of animals used for scientific purposes. The laboratory at the Institut Pasteur has the administrative authorization for animal experimentation (Permit Number 75-1554) and the protocol was approved by the Institut Pasteur Review Board that is part of the Regional Committee of Ethics of Animal Experiments of Paris Region (Permit Number: 99-174). All the invasive procedures were performed under anesthesia and all possible efforts were made to minimize animal suffering.

### Decision letter and Author response
Decision letter https://doi.org/10.7554/eLife.51247.sa1
Author response https://doi.org/10.7554/eLife.51247.sa2

## Additional files

### Supplementary files
• Supplementary file 1. Muropeptides identified by mass spectrometry. * indicates *O*-acetylated MurNAc. Acetylated muropeptides are highlighted in pink. Where multiple muropeptides coeluted as a single peak, bold text indicates the most abundant mass detected.

• Supplementary file 2. Crystallography data collection and refinement statistics of LtgA.

• Transparent reporting form

### Data availability
Coordinates and structural data have been submitted to the Protein Data Bank under the accession code 6H5F.

The following dataset was generated:

| Author(s) | Year | Dataset title | Dataset URL | Database and Identifier |
|---|---|---|---|---|
| Williams AH, Wheeler R, Hicham S, Haouz A, Taha MK, Boneca IG | 2019 | LtgA disordered Helix | http://www.rcsb.org/structure/6H5F | RCSB Protein Data Bank, 6H5F |

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
