## [Decision Letter]

**Acceptance summary:**

The surge in antimicrobial resistance, together with the emergence of new pathogenic species continues to occupy global public health authorities, all of whom have issued an urgent call for new antimicrobials with novel modes of action. In this backdrop, bacterial peptidoglycan remains a tractable area for new drug development given the proven clinical benefit of drugs that target this macromolecule. This study, which describes how an alpha helix from the peptidoglycan remodelling enzyme, LtgA, can modulate cell wall stability highlights how the careful coordination of cell wall assembly and hydrolysis can be targeted by modulating the activity of hydrolases that remodel the cell surface. The conservation of similar motifs or regulatory patterns across different species suggests that this approach can be used to identify new drug targets in several bacterial pathogens.

**Decision letter after peer review:**

[Editors’ note: the authors submitted for reconsideration following the decision after peer review. What follows is the decision letter after the first round of review.]

Thank you for submitting your work entitled "Crippling the bacterial cell wall molecular machinery" for consideration by *eLife*. Your article has been reviewed by three peer reviewers, one of whom is a member of our Board of Reviewing Editors, and the evaluation has been overseen by a Senior Editor. The reviewers have opted to remain anonymous.

Our decision has been reached after consultation between the reviewers. Based on these discussions and the individual reviews below, we regret to inform you that your work will not be considered further for publication in *eLife*.

The reviewers highlighted some interesting aspects of your work, chief among which included some novel insight on the function of LtgA, based on structure-function associations. However, there were notable concerns regards the lack of sufficient data to support the conclusions regarding the role of LtgA in coordinating peptidoglycan hydrolysis. Further, the focus of the paper did not seem to fit with the data presented.

Reviewer #1:

Summary:

Peptidoglycan (PG), a biopolymer made up of cross-linked sugars and stem peptides, is a widely distributed component of bacterial cell walls. Remodelling of this polymer requires the concerted action of a variety of PG synthesising and hydrolysing enzymes. Lytic transglycosylases are a family of PG hydrolases that break down the glycan component. These enzymes usually occur in bacteria in genetic redundancy. In their submission, Williams and colleagues describe the function of LtgA, a lytic transglycosylase, in *Neisseria meningitidis*. They build on previous findings that describe the crystal structure of this protein and describe a role for LtgA in regulating divisome function and virulence.

Key findings:

1) The description of an atomic structure of LtgA with a disordered alpha helix 30, which is involved in clamping the glycan strand during catalysis. Mutation of this helix decreases the affinity of the protein for PG.

2) Deletion of helix 30 affected growth of *N. meningitidis*, this was associated with cell division and morphological defects?

3) Removal of this helix also results in hyperacetylation of the peptidoglycan.

4) LtgA interacts with ApeI (a peptidoglycan-de-*O*-acetylase) and PBP1a (a high molecular weight penicillin binding protein).

5) LtgA and ApeI appear to enhance each other's function in a reciprocal manner.

6) Removal of helix 30 attenuated *N. meningitidis* for colonization of mice – as measured by bacteraemia in the blood – and altered induction of IL-16 and KC.

Conclusion: LtgA forms an important component of the divisome and regulates the function of other PG modifying enzymes.

Major concerns:

1) The morphological defects in Figure 1 need to be quantified, followed by a statistical comparison to wild type and the *ltgA* complement derivatives. How many cells had these observations? How many ghost cells were observed?

2) Figure 7 needs statistics for the cytokines.

3) IL-6 is proinflammatory cytokine, in most cases. The LtgA helix 30 mutant has reduced production of IL-6 (Figure 7). In this context, the statement in the Discussion (paragraph seven) that this mutant version of LtgA has pro-inflammatory potential does not make sense. The mouse phenotypes are somewhat unclear – reduced inflammation but longer persistence of the helix 30 mutant. There is a lot of conjecture in the discussion – but little evidence to substantiate the claims made.

Reviewer #2:

The manuscript entitled, "Crippling the bacterial cell wall molecular machinery". by Williams et al. describes the function of a truncation mutant in LtgA that removes alpha helix 30. While the finding may eventually turn out interesting, but at this point the inconsistencies and poor quality of the writing makes it impossible to fully understand the work. For example, the authors describe a number of proteins that bind the Δ30 protein, but not the wildtype protein, based on Figure 5. However, they go on to validate the binding of these proteins to the wildtype LtgA in Figure 6 not the mutant. They also claim that the Δ30 strain was cleared faster in mice, but they don't account for the slower growth rate of this crippled strain. There are many other problems with the work that needs to be ironed out before this could be considered for publication.

Reviewer #3:

The paper by Williams and colleagues follows up on a previous study that reported the structure of the lytic transglycosylase (LT) LtgA from *Neisseria* species. LT enzymes cut bonds in the peptidoglycan (PG) cell wall layer, but their precise role(s) in the overall process of PG biogenesis have remained elusive. Here, the authors report a variant of the LtgA structure in which an alpha-helix in the active site (H30) is disordered. It is thought that this structure may represent an intermediate in the catalytic mechanism of the enzyme. To further investigate the importance of H30 for LtgA function, a variant of the enzyme with H30 deleted (LtgA^Δ30^) is constructed and expressed in a *Neisseria* strain deleted for the native *ltgA* gene. Surprisingly, while the loss of LtgA function causes little to no phenotype, expression of LtgA^Δ30^ causes a severe growth defect and major problems with cell division/separation. Cells producing LtgA^Δ30^ are found to contain elevated levels of acetylation in the PG sugars. They are also defective in pathogenesis. Co-immunoprecipitation followed by proteomics was performed to identify interaction partners of LtgA^Δ30^ and LtgA (wild type). Although there were no significant hits for LtgA (wild type), LtgA^Δ30^ associated with other factors involved in PG biogenesis. in vitro studies were performed to show that LtgA (wild type) can interact with some of the factors identified in the proteomic study. Another factor Ape1, a PG deacetylase, was also shown to interact with LtgA (wild type) in vitro, but the interaction was not observed in the proteomic analysis. Finally, LtgA (wild type) was shown to enhance the deacetylase activity of Ape1.

From the results of these experiments the authors conclude that LtgA mediates the function of a hub of PG glycan strand modifying enzymes, orchestrates cell division, cell wall integrity, and contributes to pathogenicity. Although there are several interesting aspects to this study, most of these stated conclusions are not justified by the data. I believe the authors should carefully reconsider their interpretation of the results and consider changing the focus of the manuscript to the Ape1-LtgA connection.

My major concerns are as follows:

1) The observation that deletion of *ltgA* has no phenotype but that LtgA^Δ30^expression is detrimental to growth indicates that a defective LtgA is worse than not having it at all. One cannot conclude from this observation that LtgA orchestrates cell division, cell wall integrity, or the function of an important enzymatic complex. If this were true, then the loss of *ltgA* function would result in division and cell wall defects. The more likely explanation of the results is that the catalytically defective LtgA^Δ30^ binds glycans and/or components of the PG biogenesis machinery and somehow impairs their function. Thus, the finding that LtgA^Δ30^ expression is detrimental provides a clue to LtgA function, but more information about the mechanism behind the dominant-negative phenotype is needed to make strong statements/conclusions about LtgA's role in the cell. At present, it is hard to tell whether or not the detrimental effects of LtgA^Δ30^ production has anything to do with its normal function, especially since it is expressed at about 3-5x greater concentration than the native enzyme in the reported experiments.

2) I think it is important to know whether an active site point mutant of LtgA has a similar phenotype to LtgA^Δ30^. This will help determine whether LtgA^Δ30^ is special in some way, or if any catalytically defective enzyme will cause similar effects on cell division etc.

3) The observation that LtgA interacts with PBP1a and LtgE in vitro is interesting. However, none of the results indicate that these interactions have a functional consequence in cells or on the enzymes in vitro. Thus, the conclusion that LtgA mediates the function of a hub of PG glycan strand modifying enzymes is not an appropriate one to make without further studies.

Given that LtgA^Δ30^ expression affects PG acetylation in cells and LtgA promotes the activity of the deacetylase Ape1 in vitro, there is a much more convincing functional connection between these enzymes. I would therefore recommend rewriting the paper to focus on the dominant negative phenotype of LtgA^Δ30^ and potentially other catalytically defective LtgA variants and how this uncovered a connection with Ape1 function.

4) There is a disconnect between the co-immunoprecipitation/proteomic results and the in vitro SEC experiments. The co-IP data indicate that only LtgA^Δ30^ associates with PBP1a and other proteins, yet the SEC experiments are performed with LtgA (wild type). If the LtgA (wild type) is capable of interacting with these proteins, why did they not show up as hits in the proteomic analysis? Does LtgA^Δ30^ have a greater affinity for the interaction partners identified? This discrepancy raises concerns about the validity and reproducibility of the proteomic experiments.

[Editors’ note: further revisions were suggested prior to acceptance, as described below.]

Thank you for resubmitting your work entitled "Crippling the bacterial cell wall molecular machinery" for further consideration at *eLife*. Your revised article has been favorably evaluated by Gisela Storz (Senior Editor) and three reviewers, one of whom is a member of our Board of Reviewing Editors.

Please note that while we are asking for a substantial number of items to be addressed, we remain overall positive about the work, and are of the opinion that this all very doable within a reasonable time frame.

Summary:

The revised manuscript from Williams and colleague describes the role of the helix 30 structural motif in LtgA, a lytic transglycosylase from *Neisseria meningitidis*. Expression of a variant from of LtgA, with the helix 30 deleted, resulted in altered cell wall stability, defective cell separation, reduced virulence and hyperacetylated peptidoglycan. The authors conclude that expression of this defective version of LtgA modulates the function of the peptidoglycan de-*O*-acetylase, Ape1. Whilst some of the issue raised during the first round of review have been addressed, the following issues need attention for the manuscript to be considered further at *eLife*:

Major issues requiring new experimentation:

1). The focus on the LtgA-Ape1 connection in the revised manuscript represents a major step in the right direction. However, with this renewed focus, it becomes important to provide some data that helps explain why Ape1 activity appears to be negatively impacted by expression of the LtgA^Δ30^ mutant. At a minimum, Ape1 levels should be assessed by western blot to determine whether or not it is stably expressed in these cells.

2) Additional support for the model that Ape1 is activated by LtgA in vivo should also be provided by examining whether or not Ape1 depletion results in a morphological defect that resembles that induced by the LtgA^Δ30^ mutant.

3) Testing whether or not expression of a catalytically defective LtgA mutant induces the same phenotype as the LtgA^Δ30^ mutant is an important experiment. It will indicate whether or not there is something special about the LtgA^Δ30^ mutant or if any catalytic defect will cause the observed effects. This experiment is not a difficult one, and given its importance, is not beyond the scope of the current manuscript or a subject for further study.

4) The title of the manuscript is too vague and does not convey any useful information to a potential reader. Consider something sharper and more on-point such as, "A catalytically defective variant of a lytic transglycosylase disrupts cell morphogenesis by interfering with peptidoglycan deacetylation in Neisseria meningitidis". Or something similar to this effect.

---

## [Author Response]

[Editors’ note: the authors resubmitted a revised version of the paper for consideration. What follows is the authors’ response to the first round of review.]

Reviewer #1:[…]Major concerns:1) The morphological defects in Figure 1 need to be quantified, followed by a statistical comparison to wild type and the ltgA complement derivatives. How many cells had these observations? How many ghost cells were observed?

We thank the reviewer for this keen observation. We made the quantification of the morphological abnormalities observed by confocal microscopy. Please see Figure 3. We were unable to do the same for the scanning electron microscopy because in this panel we took high-resolution images of the individual cells or cell clusters to get a closer look at the morphological defects. The ghost cells could be observed in all the clusters we scanned. An approximation of the ratio of ghost to intact cells is 1:10.

2) Figure 7 needs statistics for the cytokines.

This has been updated. The statistical analysis was done by Kruskal-Wallis non-parametric comparison against the complemented strain with a *p*-value < 0.01. Figure 7 is now the new Figure 6 in the revised manuscript.

3) IL-6 is proinflammatory cytokine, in most cases. The LtgA helix 30 mutant has reduced production of IL-6 (Figure 7). In this context, the statement in the Discussion (paragraph seven) that this mutant version of LtgA has pro-inflammatory potential does not make sense.

We thank the reviewer for this astute observation. This sentence has been deleted in the Discussion. The paper has been refocused and the discussion has been revised.

The mouse phenotypes are somewhat unclear – reduced inflammation but longer persistence of the helix 30 mutant. There is a lot of conjecture in the discussion – but little evidence to substantiate the claims made.

We apologize if this was unclear, however, infection with the helix-30 mutant did not result in longer persistence in the blood stream but it was cleared at a faster rate. See Figure 6. We have refocused the paper and modified the Discussion.

Reviewer #2:The manuscript entitled, "Crippling the bacterial cell wall molecular machinery". by Williams et al. describes the function of a truncation mutant in LtgA that removes alpha helix 30. While the finding may eventually turn out interesting, but at this point the inconsistencies and poor quality of the writing makes it impossible to fully understand the work.

1)We thank the reviewer for the comments, but we disagree with this assessment. Based on structural knowledge we demonstrate how an LT (LtgA) can be manipulated to affect the integrity of the cell wall composition, create a defect in cell division and separation, and can be manipulated to impair the fitness of the human pathogen *Neisseria meningitidis* during infection. Our study shows that targeting LTs disrupts the function of its binding partner Ape1 whose stability and activity depends on LtgA. We demonstrate in this verified case, that besides degrading the PG, peptidoglycan degrading enzyme complexes can modulate their enzyme function and stability. These findings highlight the importance of these PG complexes in *Neisseria* biology and virulence, demonstrating their potential as future targets for drug development.

2) We are sorry that it was unclear, and we have taken significant steps to revise the manuscript both in focus and discussion. The revised paper is more focused and easier to read. It is now primarily focused on LtgA and Ape1 which has shortened the manuscript and improved the overall clarity.

For example, the authors describe a number of proteins that bind the Δ30 protein, but not the wildtype protein, based on Figure 5. However, they go on to validate the binding of these proteins to the wildtype LtgA in Figure 6 not the mutant.

We thank the reviewer for pointing this out. We apologize if this was unclear however the purpose of this experiment was misunderstood because it was not communicated effectively in our original draft of our paper. Therefore, we will explain in more detail below:

In our manuscript we used the Label-free mass spectrometry developed by Mathias Mann (PMID: 25363814). We utilized two quantification strategies: (1) spectral counting and (2) spectrometric signal intensity to measure the protein expression.

All samples were analyzed by mass spectrometry separately using the same protocol. The proteins from each sample were identified using two strategies: 1) the protein expression from each sample is estimated using the number of MS/MS spectra identifying peptide of the protein and 2) the intensity of the corresponding MS spectrum features of the protein. Our output is the list of detected proteins and the absolute or relative abundance of the proteins across all samples for each experimental run, which was done in triplicates. I would like to emphasize that the mass spectrometry can only identify enriched peptides in one sample versus another. It does not tell you whether proteins are absolutely interacting directly or whether all peptides belonging to every single protein was identified, however it can be utilized to define the molecular environment of the protein of interest. For example, in both the wild-type and the delta helix strain PBP1a, LtgE, LtgD were identified but they were more enriched in the delta helix strain. In contrast, Ape1 peptides were not identified by mass spec in either strain which could be because of the abundance of Ape1 in the bacteria or some other technical issue.

Finally, we realize that the mass spectrometry and validation experiments of the LtgA interactome broadens the story and therefore based on suggestions from reviewer 3 we have refocused the manuscripts on LtgA and Ape1. The proteomic experiments have been moved to the supplementary and its limitation and precise utility discussed briefly in the main text. We apologize again for this confusion.

They also claim that the Δ30 strain was cleared faster in mice, but they don't account for the slower growth rate of this crippled strain.

We apologize there was no detail explanation in the text. The Discussion has been modified. It now reads:

“Finally, to understand what role a defective LtgA that interferes with the normal function of the PG machinery plays in the pathogenesis of *N. meningitidis* we used a mouse infection model and showed Δ*ltgA^ltgA^*^Δ30^ strain of type *N. meningitidis* was cleared from the blood at a significantly faster rate than the wild-type, Δ*ltgA^ltgA^* or Δ*ltgA* strains. […] Additionally, the helix deleted strain is hyperacetyaled and it is known that modification of the PG makes the cell wall more susceptible to complement-mediated lysis (Zarantonelli et al., 2013; Rosain et al., 2017), and PG modification is also associated with a decreased inflammatory response (Taguchi et al.,2018).”

There are many other problems with the work that needs to be ironed out before this could be considered for publication.

We thank the reviewer for your critic. We addressed all the problems that were pointed out. The study was thorough, and comprehensive as well as cross-disciplined and while the presentation was unclear the experiments were executed with the utmost rigor in mind. As mentioned above we have focused the manuscript on LtgA and Ape1 which improved its clarity, focus and message.

Reviewer #3:[…]From the results of these experiments the authors conclude that LtgA mediates the function of a hub of PG glycan strand modifying enzymes, orchestrates cell division, cell wall integrity, and contributes to pathogenicity. Although there are several interesting aspects to this study, most of these stated conclusions are not justified by the data. I believe the authors should carefully reconsider their interpretation of the results and consider changing the focus of the manuscript to the Ape1-LtgA connection.

We agree with the reviewer that such a statement would be too preliminary, and we revised our conclusion: Please see below:

“We devised a multidisciplinary approach using structural biology to show that it is possible to target a ‘hot spot’ on an LT in order to affect bacterial growth, cell division, and cell membrane integrity, which resulted in lethal consequences for the bacteria during host infection. […] A small molecule binding to alpha helix 30 could interfere with growth and simultaneously promote bacterial clearance, mimicking the enhanced clearance of the *ltgA^ltgA^*^Δ30^ mutant in a murine infection model”.

My major concerns are as follows:1) The observation that deletion of ltgA has no phenotype but that LtgA^Δ30^ expression is detrimental to growth indicates that a defective LtgA is worse than not having it at all. One cannot conclude from this observation that LtgA orchestrates cell division, cell wall integrity, or the function of an important enzymatic complex. If this were true, then the loss of ltgA function would result in division and cell wall defects. The more likely explanation of the results is that the catalytically defective LtgA^Δ30^ binds glycans and/or components of the PG biogenesis machinery and somehow impairs their function. Thus, the finding that LtgA^Δ30^ expression is detrimental provides a clue to LtgA function, but more information about the mechanism behind the dominant-negative phenotype is needed to make strong statements/conclusions about LtgA's role in the cell. At present, it is hard to tell whether or not the detrimental effects of LtgA^Δ30^ production has anything to do with its normal function, especially since it is expressed at about 3-5x greater concentration than the native enzyme in the reported experiments.

We thank the reviewer for this observation. We have changed the sentence in the Abstract to read:

“Here we show, that the active site of LtgA can be genetically modified to, affect the integrity of the cell wall, cell division, cell separation, and can impair the fitness of the human pathogen *Neisseria meningitidis* during infection.”

We apologize if this was unclear however, LtgA^Δ30^ is produced relative to the wild type or the complemented strains at roughly 2 times greater amounts. Please see the exact quantification in Figure 2B and C. However, we agree and appreciate your point about the precision of the statement.

2) I think it is important to know whether an active site point mutant of LtgA has a similar phenotype to LtgA^Δ30^. This will help determine whether LtgA^Δ30^ is special in some way, or if any catalytically defective enzyme will cause similar effects on cell division etc.

This is a very interesting point, however the mutant we created is defective in substrate binding and therefore could affect the ability of the LtgA^Δ30^ mutant catalyze efficiently the PG. However, this would be a part of a future more extensive exploration.

*3) The observation that LtgA interacts with PBP1a and LtgE* in vitro *is interesting. However, none of the results indicate that these interactions have a functional consequence in cells or on the enzymes* in vitro*. Thus, the conclusion that LtgA mediates the function of a hub of PG glycan strand modifying enzymes is not an appropriate one to make without further studies.*

The reviewer is correct. As pointed out above in the revised version we have removed this statement from the manuscript. Additionally, primarily based on your suggestion we have further refocused the paper on Ape1 and LtgA.

*Given that LtgA^Δ30^ expression affects PG acetylation in cells and LtgA promotes the activity of the deacetylase Ape1* in vitro*, there is a much more convincing functional connection between these enzymes. I would therefore recommend rewriting the paper to focus on the dominant negative phenotype of LtgA^Δ30^ and potentially other catalytically defective LtgA variants and how this uncovered a connection with Ape1 function.*

We thank the reviewer for this very important insight, and we have revised the manuscript to focus on these two main players Ape1 and LtgA.

*4) There is a disconnect between the co-immunoprecipitation/proteomic results and the* in vitro *SEC experiments. The co-IP data indicate that only LtgA^Δ30^ associates with PBP1a and other proteins, yet the SEC experiments are performed with LtgA (wild type). If the LtgA (wild type) is capable of interacting with these proteins, why did they not show up as hits in the proteomic analysis? Does LtgA^Δ30^ have a greater affinity for the interaction partners identified? This discrepancy raises concerns about the validity and reproducibility of the proteomic experiments.*

We thank the reviewer for pointing this out. We apologize if this was unclear however, since the purpose of this experiment was largely misunderstood, because it was not communicated effectively in our original draft of our paper. We will explain below:

In our manuscript we used the Label-free mass spectrometry developed by Mathias Mann (PMID: 25363814). We utilized two quantification strategies: (1) spectral counting and (2) spectrometric signal intensity to measure the protein expression. All samples were analyzed by mass spectrometry separately using the same protocol. The proteins from each sample were identified using two strategies: 1) the protein expression from each sample is estimated using the number of MS/MS spectra identifying peptide of the protein and 2) the intensity of the corresponding MS spectrum features of the protein. Our output is the list of detected proteins and the absolute or relative abundance of the proteins across all samples for each experimental run, which was done in triplicates. I would like to emphasize that the mass spectrometry can only identify enriched peptides in one sample versus another. It does not tell you whether proteins are absolutely interacting directly or whether all peptides belonging to every single protein was identified, however it can be utilized to define the molecular environment of the protein of interest. For example, in both the wild-type and the delta helix strain PBP1a, LtgE, LtgD were identified but they were more enriched in the delta helix strain. In contrast, Ape1 peptides were not identified by mass spec in either strain which could be because of the abundance of Ape1 in the bacteria or some other technical issue.

We acknowledge that the Mass spec and validation experiments of the LtgA interactome broadens the story and therefore based on your suggestions we have further refocused the manuscripts on LtgA and Ape1. The proteomic experiments have been moved to the supplementary and its limitation and precise utility discussed briefly in the main text.

[Editors’ note: what follows is the authors’ response to the second round of review.]

Major issues requiring new experimentation:1). The focus on the LtgA-Ape1 connection in the revised manuscript represents a major step in the right direction. However, with this renewed focus, it becomes important to provide some data that helps explain why Ape1 activity appears to be negatively impacted by expression of the LtgA^Δ30^ mutant. At a minimum, Ape1 levels should be assessed by western blot to determine whether or not it is stably expressed in these cells.

We thank the reviewer for this important insight. Ape1 is produced comparatively across each strain. Please see Figure 4—figure supplement 2.

*2) Additional support for the model that Ape1 is activated by LtgA* in vivo *should also be provided by examining whether or not Ape1 depletion results in a morphological defect that resembles that induced by the LtgA^Δ30^ mutant.*

We thank the reviewers for this suggestion. In our previous study we did note that a knockout of Ape1 resulted in the increased cell size of *Neissera meningitidis (*Veyrier et al. 2013). Here we examine the *Δape1* strain for morphological defects using SEM or labeling the cell wall with FM-64. While there were no pronounced differences in the division and separation, we did find significant cell shape abnormalities and lysed bacteria i.e. irregular cell surfaces, high molecular weight blebs, asymmetrical diplococci, and lysed bacteria Please see Figure 3—figure supplement 2.

3) Testing whether or not expression of a catalytically defective LtgA mutant induces the same phenotype as the LtgA^Δ30^ mutant is an important experiment. It will indicate whether or not there is something special about the LtgA^Δ30^ mutant or if any catalytic defect will cause the observed effects. This experiment is not a difficult one, and given its importance, is not beyond the scope of the current manuscript or a subject for further study.

We thank the reviewer for the suggestion. A mutant of the catalytic residue did not result in any observable phenotype. Please see Figure 3—figure supplement 2.

4) The title of the manuscript is too vague and does not convey any useful information to a potential reader. Consider something sharper and more on-point such as, "A catalytically defective variant of a lytic transglycosylase disrupts cell morphogenesis by interfering with peptidoglycan deacetylation in Neisseria meningitidis". Or something similar to this effect.

We loved your suggestion. However, it was too long based on the character limit from eLife. So we revised it to: “A defective lytic transglycosylase disrupts cell morphogenesis by hindering peptidoglycan de-O-acetylation in meningococcus.”